# Phenotypic plasticity underlies local invasion and distant metastasis in colon cancer

Andrea Sacchetti[1†], Miriam Teeuwssen[1†], Mathijs Verhagen[1†], Rosalie Joosten[1], Tong Xu[1], Roberto Stabile[1], Berdine van der Steen[2], Martin M Watson[1‡], Alem Gusinac[1], Won Kyu Kim[3], Inge Ubink[4], Harmen JG Van de Werken[5], Arianna Fumagalli[6], Madelon Paauwe[7], Jacco Van Rheenen[8], Owen J Sansom[7,9], Onno Kranenburg[4], Riccardo Fodde[1*]

[1]Department of Pathology, Erasmus MC, Rotterdam, Netherlands; [2]Department of Otorhinolaryngology, Head and Neck Surgery, Erasmus University Medical Center, Erasmus MC, Rotterdam, Netherlands; [3]Natural Product Research Center, Korea Institute of Science and Technology, Gangneung, Republic of Korea; [4]Department of Surgical Oncology, Cancer Centre, University Medical Centre Utrecht, Utrecht, Netherlands; [5]Cancer Computational Biology Center and Department of Urology; Erasmus University Medical Center, Rotterdam, Netherlands; [6]Princess Máxima Center for Pediatric Oncology, Utrecht, Netherlands; [7]Cancer Research UK Beatson Institute, Glasgow, United Kingdom; [8]Department of Molecular Pathology, Oncode Institute, Netherlands Cancer Institute, Amsterdam, Netherlands; [9]Institute of Cancer Sciences, University of Glasgow, Glasgow, United Kingdom

*For correspondence:
r.fodde@erasmusmc.nl

[†]These authors contributed equally to this work

Present address: [‡]Gastrointestinal Translational Research Unit, Department of Gastrointestinal Surgery at Stavanger University Hospital, Stavanger, Norway

**Abstract** Phenotypic plasticity represents the most relevant hallmark of the carcinoma cell as it bestows it with the capacity of transiently altering its morphological and functional features while *en route* to the metastatic site. However, the study of phenotypic plasticity is hindered by the rarity of these events within primary lesions and by the lack of experimental models. Here, we identified a subpopulation of phenotypic plastic colon cancer cells: EpCAM[lo] cells are motile, invasive, chemo-resistant, and highly metastatic. EpCAM[lo] bulk and single-cell RNAseq analysis indicated (1) enhanced Wnt/β-catenin signaling, (2) a broad spectrum of degrees of epithelial to mesenchymal transition (EMT) activation including hybrid E/M states (partial EMT) with highly plastic features, and (3) high correlation with the CMS4 subtype, accounting for colon cancer cases with poor prognosis and a pronounced stromal component. Of note, a signature of genes specifically expressed in EpCAM[lo] cancer cells is highly predictive of overall survival in tumors other than CMS4, thus highlighting the relevance of quasi-mesenchymal tumor cells across the spectrum of colon cancers. Enhanced Wnt and the downstream EMT activation represent key events in eliciting phenotypic plasticity along the invasive front of primary colon carcinomas. Distinct sets of epithelial and mesenchymal genes define transcriptional trajectories through which state transitions arise. pEMT cells, often earmarked by the extracellular matrix glycoprotein SPARC together with nuclear ZEB1 and β-catenin along the invasive front of primary colon carcinomas, are predicted to represent the origin of these (de)differentiation routes through biologically distinct cellular states and to underlie the phenotypic plasticity of colon cancer cells.

## Introduction

Cancers of epithelial origin such as breast, prostate, pancreas, lung, and colon carcinomas are thought to develop from normal tissues through a multistep sequence of genetic events from benign precursor lesions to increasingly more malignant stages. This is exemplarily illustrated by the adenoma-carcinoma sequence in colon cancer where a stepwise buildup of genetic alterations in specific oncogenes and tumor suppressor genes underlies tumor initiation and progression (*Fearon and Vogelstein, 1990*). These alterations result in well-defined cellular changes largely reflecting the so-called 'hallmarks of cancer,' which provide different selective advantages to the developing tumor and represent essential requirements for carcinoma formation at the primary site (*Hanahan and Weinberg, 2000*). However, with regard to the capacity to disseminate through the tumor microenvironment and establish metastases in distant organ sites, epigenetic changes, rather than genetic mutations, underlie what is the most clinically relevant hallmark of cancer, namely phenotypic plasticity (*Varga and Greten, 2017*; *Teeuwssen and Fodde, 2019*).

Malignant cells, and in particular those responsible for local dissemination and distant metastasis, are often endowed with the capacity to undergo transient and reversible morphological and functional changes. In particular, epithelial to mesenchymal transition (EMT), that is, the progressive loss of epithelial features and the acquirement of a more migratory and mesenchymal phenotype (*Nieto et al., 2016*), is regarded as a crucial event in tumor cell invasion and dissemination at the primary site. EMT bestows cancer cells with stem-like plastic characteristics (*Mani et al., 2008*) needed to acquire quasi-mesenchymal features at the invasive front of the primary tumor, disseminate and attain therapy resistance, and to revert back to more epithelial states (mesenchymal to epithelial transition [MET]) at the organ site of metastasis (*Brabletz et al., 2005*). Epigenetic activation and silencing of EMT-inducing transcription factors (EMT-TFs) underlies the transient nature of these cellular modifications (*Skrypek et al., 2017*). Notwithstanding these ground rules, a very broad spectrum of molecular and cellular routes underlies EMT and the resulting phenotypic plasticity in a context-dependent fashion (*Cook and Vanderhyden, 2020*).

The 'migrating cancer stem cell' (mCSC) model has been first proposed for colon cancer by T. Brabletz (*Brabletz et al., 2005*), also as a solution to the so-called 'β-catenin paradox' (*Fodde and Brabletz, 2007*). In the majority of sporadic colorectal cancer cases, the rate-limiting loss of the *APC* tumor suppressor is predicted to lead to nuclear β-catenin translocation and full-blown Wnt signaling activation. Notwithstanding these predictions, tumor cells with nuclear β-catenin represent only a small minority of the primary lesion and tend to cluster non-randomly at the invasive front of colon carcinomas where they gain mesenchymal features to detach and disseminate into the adjacent stromal tissues. In view of these observations, it is plausible that cues secreted from the tumor microenvironment elicit EMT downstream of full-blown Wnt signaling activation, earmarked by nuclear β-catenin, in a subset of cells located at the invasive front (*Brabletz et al., 2005*; *Fodde and Brabletz, 2007*). However, the molecular and cellular mechanisms underlying Wnt and EMT activation at the invasive front of colon cancers are yet largely unknown also due to a lack of robust in vitro and in vivo models.

Previously, it was shown that human immortalized breast cancer cell lines encompass different subpopulations of cells with distinct phenotypic states and functional characteristics maintained in a dynamic equilibrium through stochastic transitions between states (*Gupta et al., 2011*). Similar observations were made in oral squamous carcinoma cell lines where distinct CSC phenotypes are present: whereas non-EMT CSCs are proliferative and retain epithelial characteristics, the EMT-competent CSC fraction is (quasi)mesenchymal and of increased cellular motility (*Biddle et al., 2011*). As such, conventional immortalized cancer cell lines may offer a valid model to elucidate the mechanisms underlying phenotypic plasticity in cancer and to identify novel EMT/CSC-related therapeutic targets.

Here, we identified and extensively characterized a subpopulation of quasi-mesenchymal colon cancer cells endowed with phenotypic plasticity that underlie local invasion and distant metastasis, and whose expression signature is predictive of reduced disease-free survival among colon cancer patients.

# Results

## Conventional colon cancer cell lines encompass a subpopulation of ZEB1-driven quasi-mesenchymal, highly metastatic, and chemo-resistant cells

To assess whether, as observed for breast cancer (*Gupta et al., 2011*), commonly employed colon cancer cell lines encompass distinct differentiated and more stem-like subpopulations of cancer cells, we first analyzed a broad panel of cell lines by FACS with antibodies directed against different (cancer) stem cell markers (CD44, CD133, CD24, ALDEFLUOR) in combination with the epithelial marker EpCAM. As shown in *Figure 1A* and *Figure 1—figure supplement 1* for HCT116 and SW480, the CD44/EpCAM combination best highlighted the presence of distinct subpopulations with a predominant CD44$^{high}$EpCAM$^{high}$ cluster and a minority of CD44$^{high}$EpCAM$^{low}$ cells. The same CD44$^{high}$EpCAM$^{low}$ cells were identified at similarly variable percentages in other commonly employed colon cancer cell lines (*Figure 1—source data 1*). Here, we mainly focused on the HCT116 and SW480 lines as they are representative of the two main colon cancer subtypes earmarked by microsatellite instability (MIN) and chromosomal instability (CIN; also known as microsatellite stable [MSS]), respectively (*Lengauer et al., 1997*). For the sake of clarity and readability, from this point on we will refer to the CD44$^{high}$EpCAM$^{low}$ and CD44$^{high}$EpCAM$^{high}$ subpopulations of colon cancer cells as EpCAM$^{lo}$ and EpCAM$^{hi}$, respectively.

To characterize the EpCAM$^{lo}$ and EpCAM$^{hi}$ colon cancer cells, they were sorted by FACS and shown to have distinct epithelial- (EpCAM$^{hi}$) and mesenchymal-like (EpCAM$^{lo}$) morphologies (*Figure 1B*). Likewise, EpCAM$^{lo}$ cells were shown to have significantly increased migratory and invasive ability when compared with their EpCAM$^{hi}$ counterparts (*Figure 1—figure supplement 2A*).

RT-qPCR analysis of the sorted subpopulations revealed differential expression of EMT-associated marker genes and transcription factors. Significantly reduced mRNA levels of *EPCAM* and E-cadherin (*CDH1*), and increased expression of vimentin (*VIM*) and of the EMT-transcription factor *ZEB1* were observed in EpCAM$^{lo}$ cells from both lines, as also confirmed by immunofluorescence (IF) analysis (*Figure 1—figure supplement 2B–D*). Accordingly, knockdown of *ZEB1* expression by shRNA dramatically decreased the percentage of EpCAM$^{lo}$ cells in both lines (*Figure 1—figure supplement 3A–C*).

Expression of *ZEB1* (and *ZEB2*) has been shown to be regulated by the miR-200 superfamily of microRNAs that target specific 3′UTR sequences (*Brabletz and Brabletz, 2010*). RT-qPCR analysis of sorted cells revealed significantly reduced expression levels of all five miR-200 members in EpCAM$^{lo}$ cells from both cell lines (*Figure 1—figure supplement 3D*), that is, in agreement with the observed increase in *ZEB1* expression. Proliferation and cell cycle analysis indicated decreased mitotic activity in EpCAM$^{lo}$ cells from both cell lines (*Figure 1—figure supplement 3E,F* and *Figure 1—source data 2*).

In view of the well-established correlation between EMT and therapy resistance (*Dean et al., 2005*), EpCAM$^{lo}$ cells were cultured in the presence of oxaliplatin and 5-fluorouracil (5-FU) and their viability compared with that of EpCAM$^{hi}$ and bulk cells by metabolic activity assay (MTT). EpCAM$^{lo}$ cells showed increased viability at all tested oxaliplatin (*Figure 1—figure supplement 4A,C*, left panels) and 5-FU (*Figure 1—figure supplement 4B,D*, left panels) concentrations. Likewise, regrowth assays revealed that EpCAM$^{lo}$ cells from both cell lines are able to re-enter the cell cycle at a broad range of oxaliplatin and 5-FU concentrations when compared with EpCAM$^{hi}$ cells (*Figure 1—figure supplement 4*, right panels).

Last, to assess in vivo their capacity to form metastatic lesions in the liver, HCT116 and SW480 bulk and EpCAM$^{hi/lo}$ sorted cells were injected in the spleen of immune-incompetent recipient mice. EpCAM$^{lo}$ cells from both lines resulted in significantly more liver metastases than with EpCAM$^{hi}$ and bulk cells (*Figure 1C,D*). Notably, immunohistochemistry (IHC) analysis of the resulting liver metastases revealed a heterogeneous pattern of intracellular β-catenin, with membranous and cytoplasmic localization in cells from within the center of the lesion, and nuclear β-catenin accumulation in cells localized in the periphery, thus recapitulating what is observed in primary colon carcinomas (*Fodde and Brabletz, 2007*; *Kirchner and Brabletz, 2000*; *Figure 1—figure supplement 5A*). FACS analysis of the EpCAM$^{lo}$-derived liver metastases revealed predominant epithelial features with a vast majority of EpCAM$^{hi}$ cells (>99%), thus highlighting their striking plasticity and the key role played by MET in metastasis formation (*Figure 1—figure supplement 5B* and *Figure 1—source data 3*).

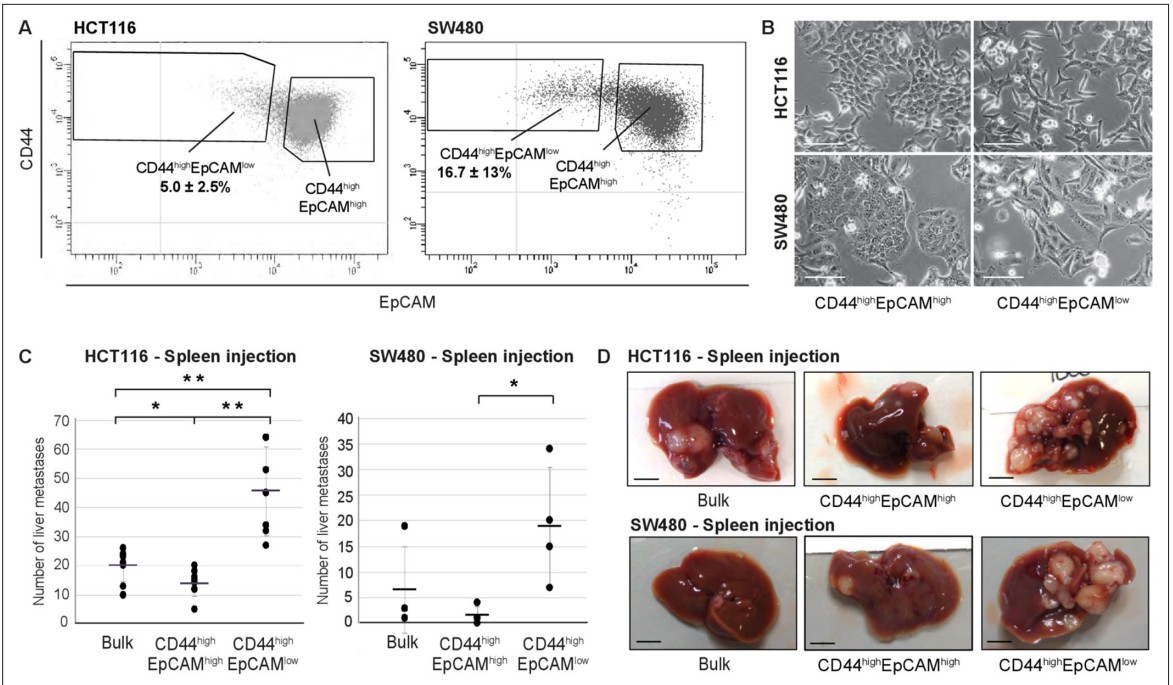

**Figure 1.** Identification and characterization of EpCAM$^{lo}$ cells in colon cancer cell lines. (A) Flow cytometric analysis of the colon cancer cell lines HCT116 (left panel) and SW480 (right panel) with antibodies directed against CD44 and EpCAM. EpCAM/CD44-positive and -negative regions (gray quadrants) were defined as in Figure supplement 1 using multiple isotype controls and are shown by the quadrants in the plots. Notably, both HCT116 and SW480 revealed a continuum of different EpCAM and CD44 expression levels with a large CD44$^{high}$EpCAM$^{high}$ (EpCAM$^{hi}$) cluster followed by a tail of gradually decreasing EpCAM and increasing CD44 levels. By applying specific gates, cells were divided in a large EpCAM$^{hi}$ cluster, together with a considerably smaller CD44$^{high}$EpCAM$^{low}$ (EpCAM$^{lo}$) subpopulation. To ensure good separation from the large EpCAM$^{hi}$ cluster and maximal sorting purity, EpCAM$^{lo}$ cells were gated as CD44$^{hi}$ events ≤ 60% of the EpCAM fluorescence intensity of the left border of the EpCAM$^{hi}$ gate and sorted from ≤50% of that value. Variable percentages of EpCAM$^{lo}$ cells were found to feature the HCT116 (5.0% ± 2.5%) and SW480 (16.7% ± 13%) cell lines, respectively. For the sake of simplicity, gates are shown in the figure only if they encompass sizeable percentages of cells. Graphs show representative analysis of one experiment. (B) Phase-contrast microscopy images of sorted EpCAM$^{hi}$ and EpCAM$^{lo}$ cells from HCT116 (upper images) and SW480 (lower images) cells. While EpCAM$^{hi}$ cells formed compact colonies with characteristic epithelial morphology, EpCAM$^{lo}$ cells showed a more spindle- and mesenchymal-like appearance. Scale bar: 100 µm. (C) Intrasplenic injection of bulk, EpCAM$^{hi}$, and EpCAM$^{lo}$ cells from HCT116 (left panel) and SW480 (right panel). For each transplantation experiment, 2 × 10$^4$ cells were injected in the spleen of a recipient NSG mouse. 4 (HCT116) and 8 (SW480) weeks after injection, mice were sacrificed and individual tumors counted. Single and double asterisks indicate significant differences (p<0.05 and p<0.01, respectively). HCT116: bulk (n = 8), EpCAM$^{hi}$ (n = 9), and EpCAM$^{lo}$ (n = 7). SW480: bulk (n = 4), EpCAM$^{hi}$ (n = 4), and EpCAM$^{lo}$ (n = 4). (D) Images of mouse livers 4 (HCT116) and 8 (SW480) weeks after orthotopic injection with 10$^4$ cells. Scale bar: 5 mm.

The online version of this article includes the following source data and figure supplement(s) for figure 1:

**Source data 1.** EpCAM$^{lo}$ cells among colon cancer cell lines.

**Source data 2.** Cell cycle analysis of EpCAM$^{hi}$ and EpCAM$^{lo}$ cells in HCT116 and SW480.

**Source data 3.** Quantification of EpCAM$^{hi/lo}$ percentages of all liver metastases as determined by FACS.

**Figure supplement 1.** Further characterization of EpCAM$^{lo}$ cells in colon cancer cell lines: FACS analysis.

**Figure supplement 2.** Further characterization of EpCAM$^{lo}$ cells in colon cancer cell lines: migration/invasion and EMT analysis.

**Figure supplement 3.** Further characterization of EpCAM$^{lo}$ cells in colon cancer cell lines: EMT and cell cycle analysis.

**Figure supplement 4.** Further characterization of EpCAM$^{lo}$ cells in colon cancer cell lines: chemoresistance.

**Figure supplement 5.** Further characterization of EpCAM$^{lo}$ cells in colon cancer cell lines: invasive and metastatic behaviour.

In order to validate the role of *ZEB1*-driven EMT in the establishment of the EpCAM$^{lo}$ subpopulation of colon cancer cells in a cell line-unrelated in vivo model mimicking the course of events observed in colon cancer patients, we employed mouse intestinal organoids carrying specific mutations at the *Apc*, *Kras*, and *Tp53* genes (*Apc$^{fl/fl}$::Kras$^{G12D/+}$::Trp53$^{fl/R172H}$*; AKP) (*Fumagalli et al., 2017*). Orthotopic transplantation of AKP organoids results in the establishment of primary carcinomas in the caecum and subsequent metastasis development at distant organ sites, mainly liver and lungs (*Fumagalli et al., 2018*). We further modified the AKP organoids by tagging them with GFP and a

click beetle luciferase (*Hall et al., 2018*) and by making them doxycycline-inducible for the expression of mouse *Zeb1* (AKP-Z) (*Figure 1—figure supplement 5C*). Upon orthotopic transplantation of the AKP-Z organoids and subsequent establishment of the primary tumor in the caecum, mice were administered doxycycline for 1 week in the drinking water to induce *Zeb1* expression. FACS analysis of the primary tumor revealed an increase in EpCAM<sup>lo</sup> cells from 4.8% in the non-induced tumors up to 22–76% upon dox-induction of *Zeb1* expression (*Figure 1—figure supplement 5D*). As expected, only a marginal increase of lung and liver metastases was observed in AKP-Z transplanted mice upon continuous dox administration in the drinking water for 8 weeks when compared with control (no dox) animals (*Figure 1—figure supplement 5E*), likely to result from the continuous induction of *Zeb1* expression and the consequent inhibition of METs essential for metastasis formation (*Brabletz et al., 2005*).

Overall, the results show the presence within colon cancer cell lines of EMT-driven, quasi-mesenchymal and therapy-resistant EpCAM<sup>lo</sup> cells with increased invasive and metastatic capacity. *ZEB1* expression underlies in vivo the establishment and maintenance of the subpopulation of EpCAM<sup>lo</sup> colon cancer cells, thereby contributing to increased dissemination along the invasion-metastasis cascade.

## EpCAM<sup>lo</sup> colon cancer cells are maintained in equilibrium with EpCAM<sup>hi</sup> through stochastic state transitions

To further investigate the plasticity of EpCAM<sup>lo</sup> cancer cells, we assessed their capacity to differentiate into the more epithelial EpCAM<sup>hi</sup> type and reconstitute the heterogeneous composition of the parental cell lines. Sorted EpCAM<sup>lo/hi</sup> cells from HCT116 and SW480 were grown separately under conventional culture conditions and analyzed by FACS at different time points. As shown in *Figure 2A*, while the majority of EpCAM<sup>lo</sup> cells from both cell lines revert to the epithelial phenotype, only a minority of EpCAM<sup>hi</sup> cells switches to the more mesenchymal state. Accordingly, *CDH1* and *EPCAM* expression was significantly increased in 'late' (e.g., cultured for >60 and >100 days, respectively) vs. 'early' EpCAM<sup>lo</sup> cells (e.g., collected after <7 days of culture), whereas *VIM* and *ZEB1* expression was decreased in 'late' vs. 'early' EpCAM<sup>lo</sup> cells (*Figure 2—figure supplement 1A*). In addition, the migratory capacity of 'late' EpCAM<sup>lo</sup> HCT116 cells was reduced to levels comparable with those of 'early' EpCAM<sup>hi</sup> cells (*Figure 2—figure supplement 1B*).

To exclude cross-contamination between subpopulations, single EpCAM<sup>hi/lo</sup> cells from both cell lines were sorted into 96-well dishes, cultured up to 70–80 days, and analyzed by FACS at intermediate and end time points. As shown in *Figure 2B* and *Figure 2—source data 1*, the majority of EpCAM<sup>lo</sup> single cells were capable of generating substantial percentages of EpCAM<sup>hi</sup> progeny to eventually recapitulate the heterogeneous composition of the parental cell lines (e.g., *Figure 2B*, clones 2F12 and 2E10). A minority of the cells, however, appears to have lost this plasticity and retains, even after extended culture, the EpCAM<sup>lo</sup> phenotype (e.g., *Figure 2B*, clone 2H12). In contrast, the majority of EpCAM<sup>hi</sup> single cells retained their epithelial features with <1% switching to the EpCAM<sup>lo</sup> state (*Figure 2B* and *Figure 2—source data 1*).

Based on these results, a two-state Markov model was developed to estimate the average probabilities to transition from one state to the other. First, the FACS data was employed to estimate the average doubling time of cells in both populations; slightly increased doubling times were reported for EpCAM<sup>lo</sup> compared to EpCAM<sup>hi</sup> cells in both lines (HCT116: 1.09 vs. 1.00 days; SW480: 1.86 vs. 1.68 days). Next, we employed least-square optimization to estimate the transition probabilities that best fit the observed population dynamics. The fitted model predicts that both subpopulations have a high probability to retain their cell identity, with minor though significant likelihood to transit to the other state. Of note, EpCAM<sup>lo</sup> cells show a higher transition probability compared to EpCAM<sup>hi</sup> (8.1- and 4.0-fold in HCT116 and SW480, respectively) (*Figure 2C*).

Due to the observed differences in doubling times between the two states, subclones with a lower EpCAM<sup>hi>lo</sup> transition probability will experience a slight growth advantage, which will become prevalent in the long run. We ran a simulation of this effect by starting from a culture with multiple subclones having distinct transition probabilities; the results indicate that subclones with lower plasticity gain dominance within a few months (*Figure 2—figure supplement 2*). Consequently, especially in the long run, the percentage of EpCAM<sup>lo</sup> cells will decrease as observed in late cultures (*Figure 1—source data 1*).

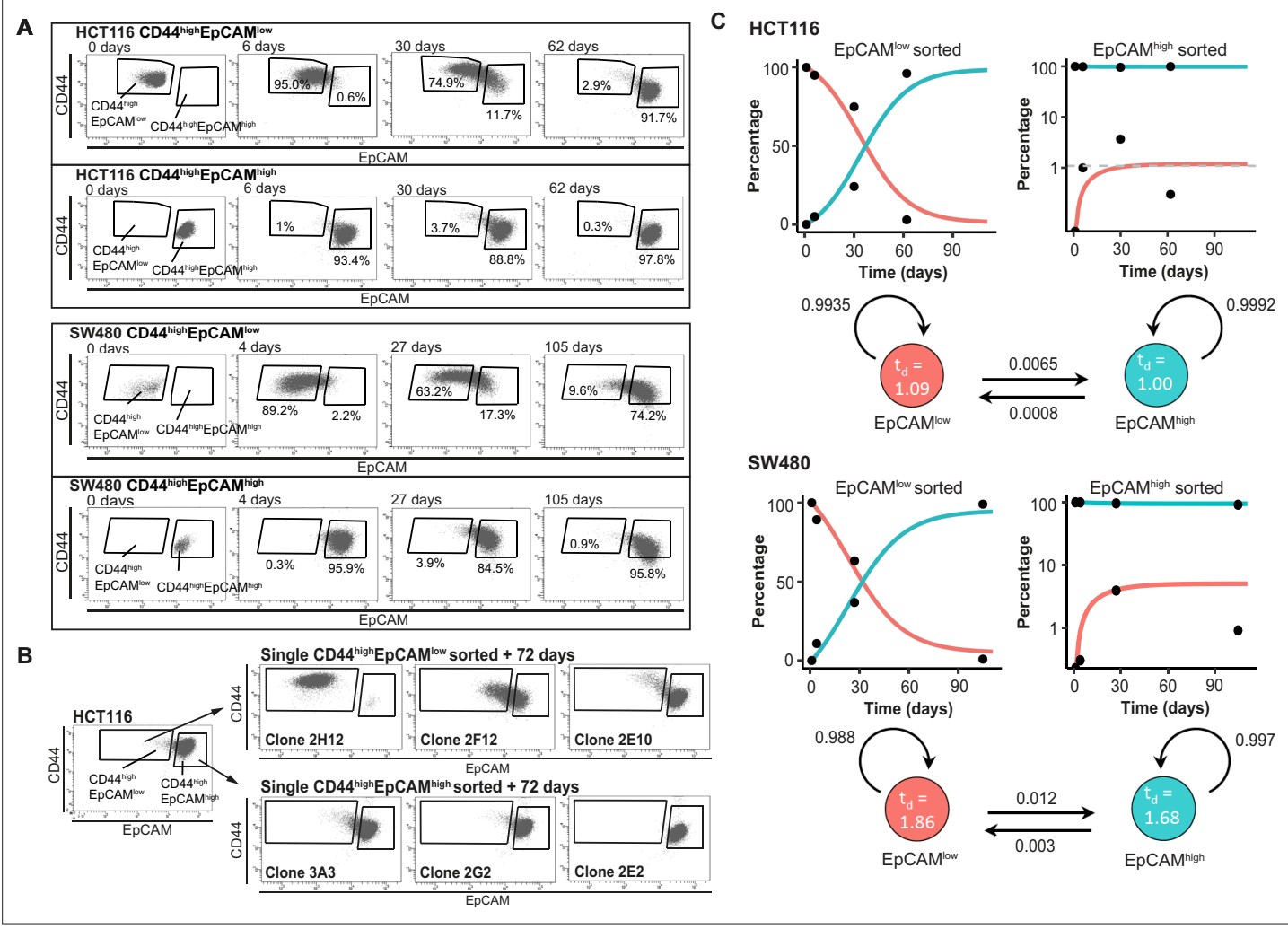

**Figure 2.** Phenotypic plasticity maintains EpCAM$^{lo}$ and EpCAM$^{hi}$ cells in a stochastic equilibrium. (**A**) Analysis of plasticity of EpCAM$^{hi}$ and EpCAM$^{lo}$ cells from HCT116 (upper panel) and SW480 (lower panel). EpCAM$^{hi}$ and EpCAM$^{lo}$ cell fractions were sorted and plated in culture. At different time points, as indicated, cells were reanalyzed by flow cytometry for their levels of CD44 (*y*-axis) and EpCAM (*x*-axis) expression. (**B**) Flow cytometric analysis of single CD44$^{hi}$EpCAM$^{hi}$ and CD44$^{hi}$EpCAM$^{lo}$ HCT116 cells sorted by FACS and cultured for 72 days. Three representative individual single-cell clones per cell fraction are shown. (**C**) Dynamics of the EpCAM$^{hi}$ and EpCAM$^{lo}$ subpopulations from the HCT and SW480 cell lines as measured by FACS (% of total) over time. Under each graph, a schematic shows the estimated transition probabilities from the fitted two-state Markov model.

The online version of this article includes the following source data and figure supplement(s) for figure 2:

**Source data 1.** Phenotypic plasticity maintains EpCAM$^{lo}$ and EpCAM$^{hi}$ cells in a stochastic equilibrium: clonal analysis.

**Figure supplement 1.** Further characterization of phenotypic plasticity in EpCAM$^{lo}$ and EpCAM$^{hi}$ cells.

**Figure supplement 2.** Simulation of the HCT116 two-state Markov model with a non-homogenous starting population.

Overall, the above results highlight the high plasticity and stem-like features of EpCAM$^{lo}$ cells in their ability to acquire epithelial features and reconstitute the heterogeneous composition of the parental cell line, independently of external factors other than the conventional culture conditions here employed.

## Enhanced Wnt signaling activation underlies EMT and the establishment of EpCAM$^{lo}$ colon cancer cells

In order to elucidate the mechanisms underlying plasticity and EMT in EpCAM$^{lo}$ colon cancer cells, RNAseq analysis was performed on the sorted subpopulations from the HCT116 and SW480 lines. Multidimensional scaling (MDS) showed a separation in the second dimension of EpCAM$^{hi}$ and EpCAM$^{lo}$

cells in both cell lines (*Figure 3A*). A total of 152 and 353 differentially regulated genes were identified between the EpCAM$^{hi}$ and EpCAM$^{lo}$ cells in HCT116 and SW480, respectively (p. adjusted<0.01, when applying a log$_2$ fold change <−1.5 and > 1.5). Among these, only a relatively few (n = 34) were common to both lines (*Figure 3—source data 1*). However, Ingenuity Pathway Analysis (IPA) revealed that genes differentially expressed in each cell line reflect common molecular and cellular functions including cellular motility, cellular assembly and organization, and drug metabolism (*Figure 3—figure supplement 1*). Nonetheless, IPA of the combined EpCAM$^{lo}$ expression profiles highlighted significant associations with EMT regulation, Wnt/β-catenin signaling, human ESC pluripotency, IL8 signaling, and colorectal cancer metastasis (*Figure 3B* and *Figure 3—figure supplement 1*).

The activation of canonical Wnt signaling in EpCAM$^{lo}$ colon cancer cells is of interest in view of the fact that both cell lines harbor mutations (loss and gain of APC and β-catenin function in SW480 [*Nishisho et al., 1991*] and HCT116 [*Ilyas et al., 1997*], respectively) predicted to result in the constitutive activation of the pathway. Notwithstanding the latter, Wnt appears to be increased in EpCAM$^{lo}$ cells, possibly due to the epigenetic activation/inhibition of synergistic/antagonistic loci. In view of these observations and of the established functional link between Wnt and EMT (*Lamouille et al., 2014*; *Ghahhari and Babashah, 2015*), we evaluated whether Wnt signaling 'super-activation' in the already Wnt-ON HCT116 and SW480 cell lines could expand the relative size of the EpCAM$^{lo}$ subpopulations. Indeed, upon treatment with the glycogen synthase 3β (GSK3β) inhibitor CHIR99021 (Chiron), a robust Wnt signaling activation was observed in both cell lines by TopFLASH reporter assay (*Figure 3C*). FACS analysis of the treated cell lines showed that the enhancement of Wnt signaling led to an approximately threefold increase of the EpCAM$^{lo}$ population in the HCT116 cell line, though not in SW480 (*Figure 3D*). However, IF analysis showed that Chiron treatment was consistently accompanied by an increase in *ZEB1* expression in both cell lines, in agreement with the role of *ZEB1* as a downstream Wnt target (*Sánchez-Tilló et al., 2011*; *Sánchez-Tilló et al., 2015*; *Figure 3E*).

To further validate the role of Wnt in the establishment of the EpCAM$^{lo}$ subpopulation, we took advantage of the recently generated isogenic subclones of the HCT116 cell line with distinct β-catenin genotypes, namely wild type (HCT116-WT), hetero- (HCT116-P), and homozygous (HCT116-MT) for the Ser45del mutant allele (*Kim et al., 2019*). FACS analysis of these cell lines revealed a progressive increase in the EpCAM$^{lo}$ subpopulation from 3.9% in HCT116-WT to 7.9% in HCT116-P and 28.7% in HCT116-MT cells (*Figure 3F*). The observed increase in EpCAM$^{lo}$ cells in the isogenic HCT116 lines matches their morphological features ranging from a distinct epithelial-like morphology in HCT116-WT to a progressively increased spindle-shaped and scattered patterns in HCT116-P and -MT cells, as originally reported by *Kim et al., 2019*.

In the majority of colon cancers, nuclear accumulation of β-catenin is exclusively observed at the invasive front where tumor cells are more exposed to growth factors and cytokines from the stromal microenvironment likely to further enhance Wnt signaling in a localized fashion, thus triggering EMT, invasion, and distant metastasis (*Brabletz et al., 2005*; *Fodde and Brabletz, 2007*). We analyzed the invasive front of a small cohort of colon carcinomas by IHC with antibodies directed against β-catenin and ZEB1 in consecutive sections. As shown in *Figure 3—figure supplement 2*, co-localization of nuclear β-catenin and ZEB1 expression was found in 5 out of 25 cases investigated.

Overall, the results highlight the key role played by enhanced Wnt signaling activation in establishing and maintaining the EpCAM$^{lo}$ subpopulation of colon cancer cells through *ZEB1* upregulation and EMT induction.

## EpCAM$^{lo}$ cells are associated with the CMS4 group of patients with shorter disease-free and overall survival

Distinct recurrent gene expression patterns underlie the recently proposed classification of human colon cancers in four consensus molecular subtypes (CMS1–4) (*Guinney et al., 2015*). Of these, the mesenchymal CMS4 subtype has the greatest propensity to form metastases. While fibrosis is a hallmark of CMS4 and a dominant contributor of mesenchymal gene expression, the cancer cells themselves can also express genes reflecting a (quasi-)mesenchymal state. Accordingly, the CMS4 subtype was identified in tumor-derived organoids and cell lines, suggesting that CMS4 is an intrinsic property of the mesenchymal colon cancer cell (*Vellinga et al., 2016*). Therefore, we asked whether expression of the signatures derived from the RNAseq analysis of EpCAM$^{lo}$ cells would correlate with the CMS classification of human colon cancers and cell lines. To this end, we employed a compiled dataset

 Research article

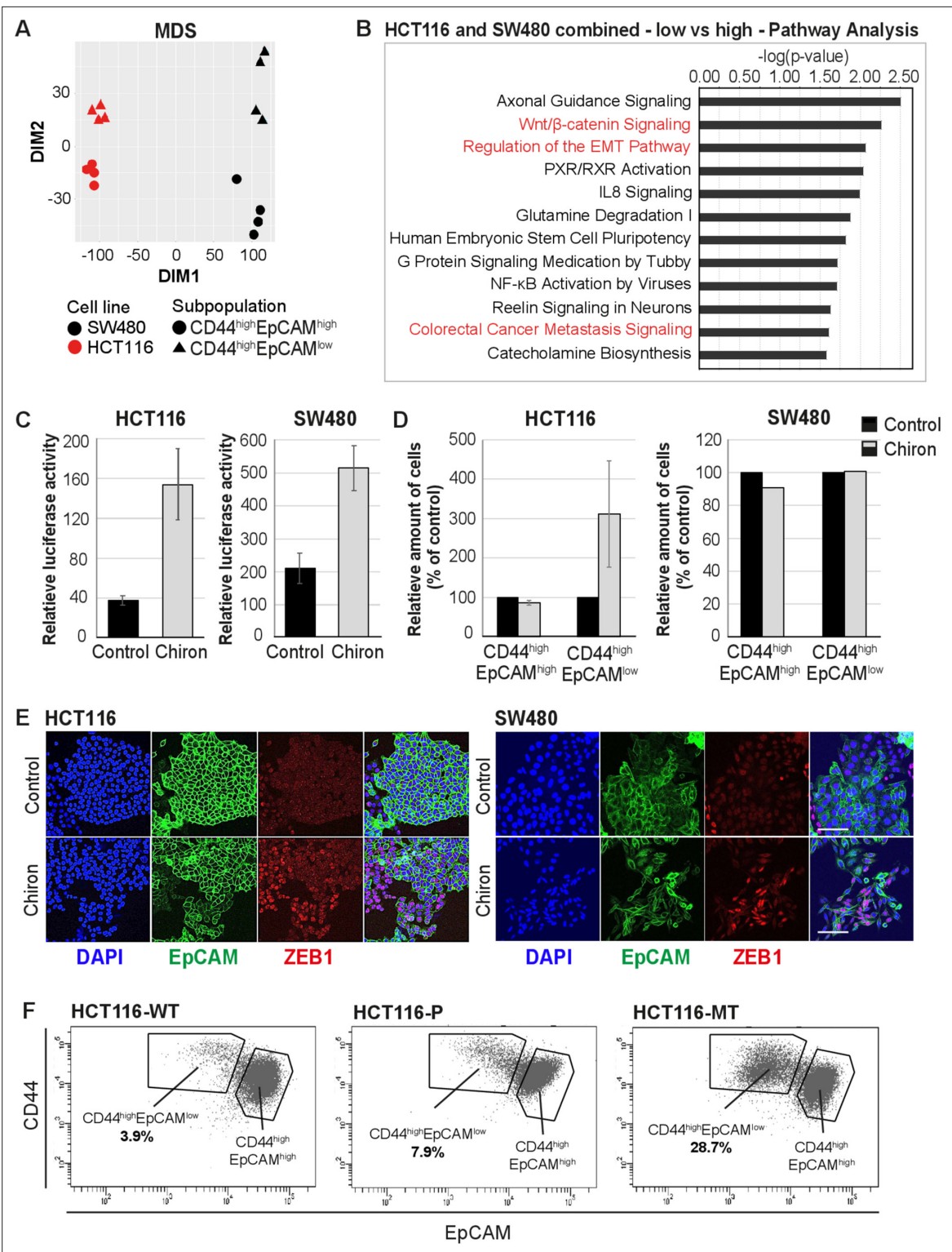

**Figure 3.** RNAseq analysis reveals enhanced Wnt signaling in EpCAM[lo] cells. (**A**) Multidimensional scaling analysis of RNAseq profiles of EpCAM[hi] and EpCAM[lo] cells from the HCT116 and SW480 lines. Red: HCT116, black: SW480, circle: EpCAM[hi], triangle: EpCAM[lo]. (**B**) Ingenuity Pathway Analysis (IPA) of the HCT116 and SW480 expression profiles from the multicell line analysis (p adjusted value <0.01; $\log_2$ fold change <–1.5 and >1.5). Red marked pathways highlight the enhanced involvement of pathways involved in epithelial to mesenchymal transition, Wnt signaling, and the formation of colon cancer metastasis in the EpCAM[lo] subpopulation compared to EpCAM[hi] cells. (**C**) TOP-Flash luciferase reporter analysis of Wnt signaling activity in colon cancer cell lines HCT116 and SW480 upon treatment with 4 µM Chiron for 3 days. Each bar represents the mean ± SD of two independent experiments.

*Figure 3 continued on next page*

Figure 3 continued

(**D**) Flow cytometric analysis using antibodies directed against CD44 and EpCAM of control and 4 μM Chiron—figure supplemented HCT116 (**A**) and SW480 (**B**) cultures. Graphs show percentage of cells within the CD44hiEpCAMhi and CD44hiEpCAMlo gates relative to the control. Each bar represents the mean ± SD of two independent experiments. (**E**) Immunofluorescence analysis of control and Chiron-treated HCT116 (left panel) and SW480 (right panel) cells. After 3 days of treatment, cells were fixed with 4% paraformaldehyde and stained with antibodies against EpCAM (green) and ZEB1 (red). Nuclei were visualized by DAPI staining of DNA (blue). Scale bar: 100 μm. (**F**) Flow cytometric analysis of three HCT116 cell lines with differential β-catenin mutation status, a parental HCT116 (HCT116-P) cell line harboring one WT and one mutant allele (Ser45 del), and two HCT116-WT and HCT116-MT cell lines harboring one WT or one mutant allele, respectively, generated by disruption of the other allele in the parental cell line (*Kim et al., 2019*).

The online version of this article includes the following source data and figure supplement(s) for figure 3:

**Source data 1.** List of differentially expressed genes in EpCAMlo vs. EpCAMhi cells in HCT116 and SW480.

**Figure supplement 1.** Gene ontology (GO) of molecular and cellular functions in HCT116 (upper panel), SW480 (middle), and the combined (bottom panel) gene lists.

**Figure supplement 2.** Hematoxylin and eosin (H&E) and immunohistochemistry (IHC) analyses with antibodies directed against beta-catenin and ZEB1 in consecutive sections of colon cancers from three unrelated patients with sporadic colon cancer.

encompassing expression data relative to 3232 human colon cancers classified as CMS1–4 (*Guinney et al., 2015*). Expression of the HCT116 and SW480 signatures was highly correlated with each other (*Figure 4A*), with the CMS4 signature genes (n = 143) (*Figure 4A–B*), and with the expression signature of colon cancer cell lines previously classified as mesenchymal-like (CCS3) (*De Sousa et al., 2013*; *Figure 4—figure supplement 1A*).

Next, we used both HCT116 and SW480 signatures from the bulk RNAseq to generate high, intermediate, and low expression subgroups in the CMS cohort through k-means clustering (*Figure 4— figure supplement 1B*). High expression of the EpCAMlo signatures from both cell lines identifies a group of primary colon cancer patients with a high propensity to form metastases and with significantly shorter overall survival (*Figure 4C*).

Overall, the above data strongly link the expression of the EpCAMlo signatures derived from common colon cancer cell lines to the CMS4 subtype and to shorter disease-free and overall survival.

## scRNAseq analysis of EpCAMlo colon cancer cells reveals high heterogeneity and partial EMT intermediate stages

To further elucidate the heterogeneity and molecular mechanisms underlying the phenotypic plasticity of EpCAMlo colon cancer cells, single-cell RNA sequencing (scRNAseq) was performed using chromium controller (10 X Genomics) technology on sorted subpopulations from HCT116 and SW480. More than 1000 cells were analyzed from each subpopulation and sequenced to a depth of approximately 50,000 reads each with the MiSeq System (Illumina). After dimension reduction with tSNE, the EpCAMlo and EpCAMhi cells clustered in separate groups in HCT116 (*Figure 5A*), whereas the SW480 cells showed a partial overlap between the two subpopulations and a distinct EpCAMhi cluster, identified as the non-adherent subpopulation within the SW480 cell line ('spheres') (*Hirsch et al., 2014*; *Yi et al., 2020*; *Figure 5—figure supplement 1A*). Substantial overlap between the EpCAMhi/lo subpopulations was retained upon subsequent exclusion of the non-adherent cells in SW480, attributed to additional variance in genes independent of EpCAMhi/lo differences (*Figure 5—figure supplement 1B*). Next, the dimension reduction was repeated in supervised fashion by taking advantage of the publicly available EMT gene list from the Nanostring nCounter PanCancer Progression Panel (n = 180; 107 of which were found to overlap with the scRNAseq data) (*Figure 5—source data 1*) . Using this 'EMT signature,' the two subpopulations were clearly resolved both in HCT116 and SW480 (*Figure 5B*).

To rank and order the cells along the E-M axis, epithelial (E; n = 51) and mesenchymal (M; n = 56) gene sets were defined from the EMT signature based on the provided annotations (Nanostring) and evaluated by Gene Set Variation Analysis (GSVA). Next, an EMT score was computed by subtracting the E score from the M score (EMT = M – E), while a partial EMT score was obtained based on the co-expression of the E and M gene sets (pEMT = min[E,M]). The expression of genes previously found to be differentially expressed between EpCAMhi/lo by bulk RNAseq was then evaluated over the course of EMT progression (*Figure 5C*). While HCT116 showed a clear transition from EpCAMhi to EpCAMlo with a minority of 'in-between' cells earmarked by high pEMT scores, in SW480 the transition appeared to be accompanied by higher levels of intercellular variance. We continued to use

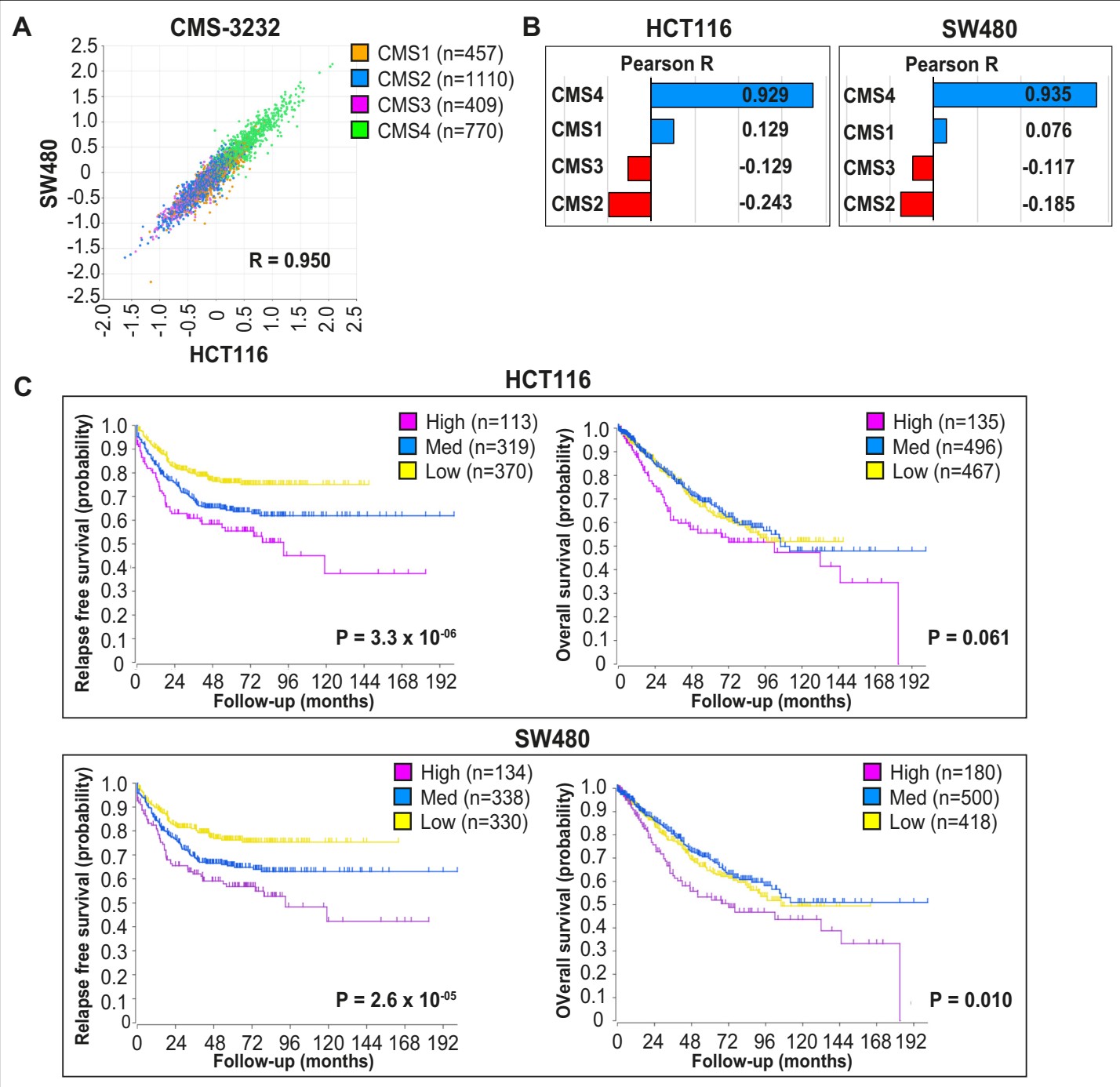

**Figure 4.** EpCAM[lo] gene expression profiles correlate with CMS4 colon cancer patients with shorter disease-free and overall survival. (**A**) Correlation of meta-gene expression values of the signatures derived from EpCAM[lo] HCT116 and SW480 cells in the consensus molecular subtype (CMS)3232 human colon cancer cohort (*Guinney et al., 2015*). (**B**) Correlation of meta-gene expression values of the signatures derived from EpCAM[lo] HCT116 and SW480 cells with expression of CMS classifier genes positively identifying each of the four molecular subtypes. (**C**) Kaplan– Meier analysis. The gene sets identifying the EpCAM[lo] cells from both HCT116 and SW480 cell lines were used to cluster the tumors in the CMS3232 cohort into high (purple), intermediate (blue), and low (yellow) expression groups by k-means clustering. The Kaplan–Meier method was subsequently used to assess significant differences in relapse-free (left panels) and overall (right panels) survival between the generated subgroups.

The online version of this article includes the following figure supplement(s) for figure 4:

**Figure supplement 1.** EpCAM[lo] expression signatures correlate with CCS3 cell lines and CMS4 colon cancer patients.

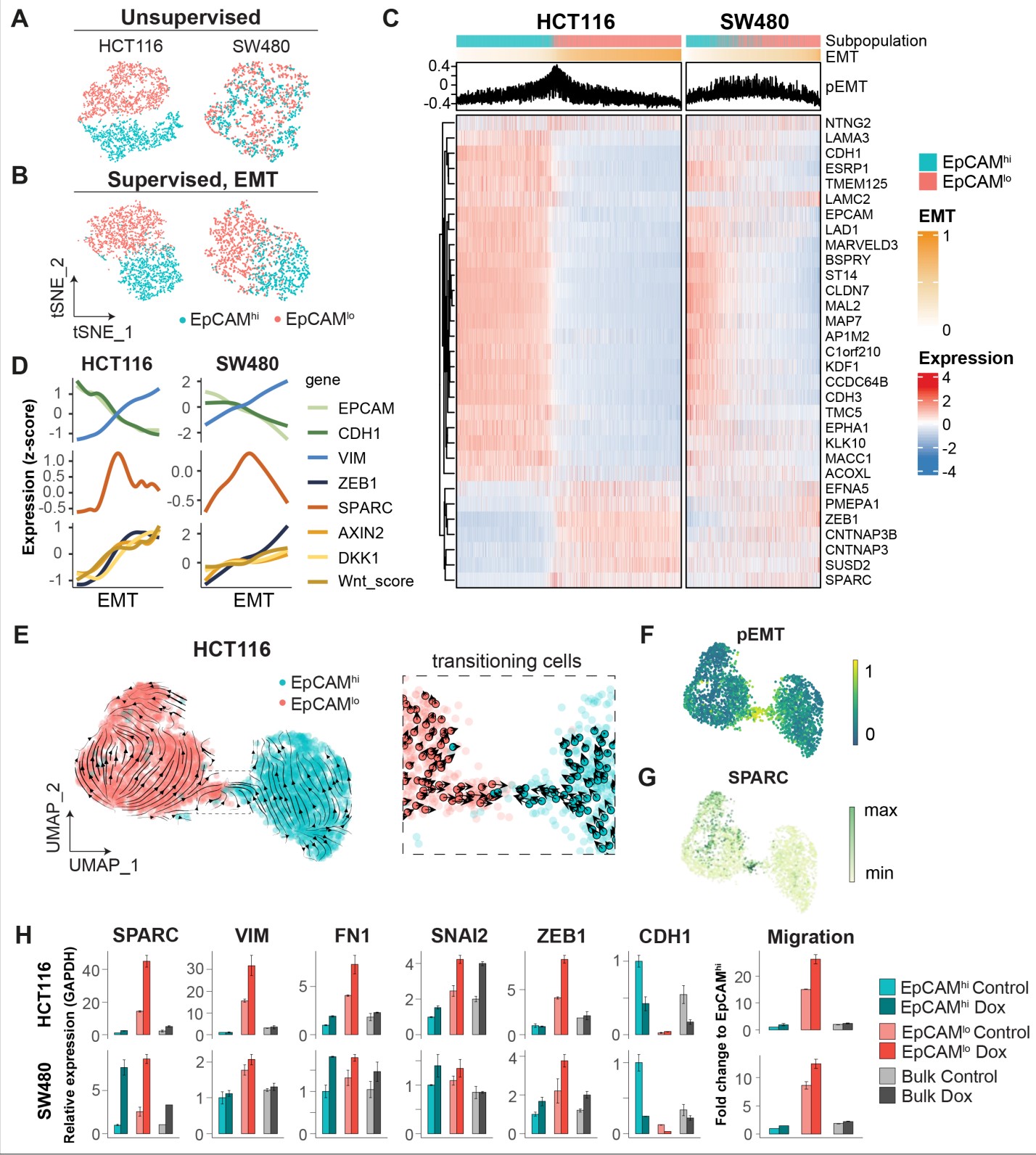

**Figure 5.** scRNAseq analysis of EpCAM[lo] cells reveals specific markers of partial EMT cells. (**A**) tSNE of HCT116 and SW480 cells based on the variable expressed genes across EpCAM[lo] and EpCAM[hi] populations. (**B**) tSNE of HCT116 and SW480 cells based on genes from the epithelial to mesenchymal transition (EMT) signature (N = 107). (**C**) Heatmap of differentially expressed genes between EpCAM[lo] and EpCAM[hi] populations. Cells were ranked according to their EMT score (EMT = M – E). (**D**) Gene expression trends projected over the EMT axis. Expression values were imputed with MAGIC,

*Figure 5 continued on next page*

*Figure 5 continued*

scaled by their z-score, and smoothened by general additive models to visualize the gene expression trend. (**E**) RNA velocity analysis of the HCT116 scRNAseq data. Cells from both populations were moving in their respective state (left panel) with a minority population of transitioning cells (right panel). (**F**) Projection of the pEMT score on the UMAP embedding of the HCT116 cell line. (**G**) The expression of SPARC on the UMAP embedding of the HCT116 cell line. (**H**) Left panels: qPCR analysis of SPARC overexpression in the subpopulations of HCT116 and SW480. Gene expression values are relative to GAPDH and normalized to the EpCAM$^{hi}$ subpopulation. Right panel: quantification of the transwell migration assay upon overexpression of SPARC.

The online version of this article includes the following source data and figure supplement(s) for figure 5:

**Source data 1.** EMT and CMS4 classifiers.

**Figure supplement 1.** Further scRNAseq analysis of EpCAM$^{lo}$ cells: EMT/MET transcriptional trajectory.

the EMT axis in both cell lines to study gene expression trends. As expected, epithelial expression of *EPCAM* and *CDH1* gradually decreased over the EMT axis, while mesenchymal genes such as *ZEB1* and *VIM*, and Wnt target genes such as *AXIN2* and *DKK1*, showed opposing behavior. More generically, the level of Wnt signaling ('Wnt.score'), measured by the activation of the WNT_PATHWAY from the PID database, increased over the EMT axis (*Figure 5D*). We evaluated the expression trends of other genes upregulated in EpCAM$^{lo}$ cells and observed that *SPARC* (osteonectin) peaked in between the extremities of the EMT axis, indicative of a partial EMT state (*Figure 5D*).

To further characterize the transition process between EpCAM$^{lo}$ and EpCAM$^{hi}$, we analyzed the HCT116 scRNAseq data by RNA velocity, an algorithm that predicts the future cell state of individual cells on a short-term timescale (usually hours), based on the ratio between spliced and unspliced mRNA. Consistent with the fitted Markov model, the majority of cells were moving in their respective state, and both states included source and sink points that could elicit or inhibit plasticity (*Figure 2— figure supplement 1C*). On a population level, EpCAM$^{lo}$ cells were more likely to transit to EpCAM$^{hi}$ cells than vice versa (by partition-based graph abstraction or PAGA; *Figure 2—figure supplement 1D*). Within the EpCAM$^{lo}$ subpopulation, *CD44* and mesenchymal genes such as *VIM* and *ZEB1* were expressed with different dynamics (*Figure 5—figure supplement 1C*). In fact, only a small population of cells was captured in the process of transitioning to the other identity (*Figure 5E*). This apparently plastic population, earmarked by the opposing velocity arrows, showed the highest pEMT score (*Figure 5F*), as well as high *SPARC* expression (*Figure 5G*).

Lastly, we performed cluster analysis on the HCT116 scRNAseq data. Unsupervised clustering using shared neighbor (SSN) modularity optimization revealed the presence of distinct subclusters (n = 8): three of EpCAM$^{hi}$ origin and five EpCAM$^{lo}$ (*Figure 5—figure supplement 1D*). To further investigate the gene expression patterns underlying the different EMT states, we performed a cluster analysis where the EMT signature genes are grouped with k-means according to their average expression in the unsupervised clusters (*Figure 5—figure supplement 1E*). This analysis revealed four sets of genes expressed in different combinations throughout the different clusters. Notably, gene set mes1, including *ZEB1*, *VIM*, and *SNAI2*, is expressed throughout the EpCAM$^{lo}$ clusters, while gene set mes2, including *SPARC*, *FN1*, and *TWIST1*, is mostly expressed in the pEMT EpCAM$^{lo}$ cluster (#7). Of note, the partial EMT clusters #6 and #7 showed distinct expression of these gene sets indicating alternative activation of specific arrays through which partial EMT cells arise (*Figure 5—figure supplement 1F*).

As *SPARC* earmarks pEMT states (*Figure 5D*), we overexpressed it in the HCT116 and SW480 cell lines and observed a marked increase in the expression of EMT-TFs (*ZEB1* and *SNAI2*) and the corresponding up- and downregulation of mesenchymal and epithelial markers in EpCAM$^{lo}$ cells and a corresponding increase in their motility and invasive capacity (*Figure 5H*).

Taken together, our in silico analysis shows substantial heterogeneity within the EpCAM$^{lo}$ subpopulation across colon cancer cell lines. Among EpCAM$^{lo}$ cells, a minority exhibits partial EMT and underlies the stochastic EpCAM$^{lo->hi}$ and EpCAM$^{hi->lo}$ transitions. Of note, pEMT is shown here not only as an '*in-between*' state earmarked by the co-expression of E- and M-specific genes, but also by specific genes like *SPARC* whose expression peaks at pEMT states.

## Identification of EpCAM^lo colon cancer cells in primary colorectal tumors

As shown above, the EpCAM^lo gene signatures derived from both cell lines show high concordance with the CMS4 subtype and are strongly associated with poor survival. Previously, it has been questioned to what extent the CMS4 profile reflects a *bona fide* clinical identity rather than being a representation of contamination from the tumor microenvironment (*Calon et al., 2015*; *Isella et al., 2015*). We addressed this issue by evaluating the CMS4 signature (here referred to as '*CMS4_RF*') (*Figure 5—source data 1*) originally developed by *Guinney et al., 2015* and by taking advantage of a recent scRNAseq study on colon cancer resections from 29 patients (*Lee et al., 2020*). Based on the latter study, as depicted in *Figure 6A*, the CMS4_RF signature shows the highest association with normal and tumor stromal cells. At the bulk RNAseq level, using the larger (n = 3232 patients) cohort of colon tumors from *Guinney et al., 2015*, the CMS4_RF signature is clearly enriched among the CMS4 tumors (*Figure 6B*) and is likewise associated with poor survival (*Figure 6C*). Hence, the CMS4_RF signature reflects the presence of tumor-associated stromal cells, likely to be enriched in patients with decreased overall survival. To question whether the presence of a subset of *bona fide* tumor cells in quasi-mesenchymal state (EpCAM^lo) may also represent a feature of colon cancers with poor prognosis, we derived a distinct signature ('*CMS4_TC*') (*Figure 5—source data 1*) by selecting genes correlated (Pearson > 0.3) with the CMS4_RF signature within the tumor epithelial fraction from the *Lee et al., 2020* study. Accordingly, the CMS4_TC signature shows the highest association with epithelial tumor cells (*Figure 6D*). Of note, at the bulk RNA level, the CMS4_TC signature shows increased association with both CMS1 and CMS4 tumors (*Figure 6E*) and outperforms the CMS4_RF signature in stratifying patients on survival (*Figure 6F*). The same CMS4_TC signature can clearly stratify patients from CMS groups other than CMS4, and in particular CMS1 and CMS3, on survival (*Figure 6—figure supplement 1*).

Finally, in order to identify EpCAM^lo cells in primary colorectal tumors, we took advantage of the scRNAseq data from *Lee et al., 2020* to annotate the tumor cells with highest association to the CMS4_TC signature as mesenchymal-like (Mes-like = 9.7% of all tumor cells). In agreement with the observations made at the bulk RNA level, tumor cells from CMS1 and CMS4 patients showed the highest association to the CMS4_TC signature (*Figure 6G*). Differential expression between the mesenchymal-like and bulk tumor cells revealed lower expression of *EPCAM*, while higher expression of *CD44* and mesenchymal markers such as *VIM*, *COL1A1*, *MMP7*, *ECM1*, and *SPARC* (*Figure 6H*), the latter previously identified to peak at intermediate EMT levels and predicted by RNAvelocity to earmark a subpopulation of plastic cells transitioning in between states (*Figure 5D–G*).

## Discussion

The progression from primary tumor to metastasis still represents an unsolved puzzle as genomic and gene expression profiles of primary cancers and their matched metastases are in general strikingly similar (*Bernards and Weinberg, 2002*). Colon cancer provides a unique example for this conundrum: although it is well-established that a sequence of gene mutations underlies the adenoma-carcinoma sequence at the primary site, no alterations have been identified in genes potentially able to underlie local invasion and distant metastasis. Hence, the capacity to metastasize could already be predetermined by specific mutations acquired by tumor cells at relatively early stages of tumorigenesis. However, this does not yet explain why and how only rare cells within the primary lesion acquire a metastatic phenotype that endows them with increased cell motility and with the capacity to invade the stromal microenvironment to eventually home at distant organ sites and form metastases. From this perspective, phenotypic plasticity likely represents a relevant mechanism for metastasizing colon carcinoma cells to transiently and reversibly change their cellular identity along the invasion-metastasis cascade (*Varga and Greten, 2017*; *Teeuwssen and Fodde, 2019*).

The vast majority of colon cancers are initiated by mutations at the *APC* gene predicted to lead to constitutive Wnt signaling by β-catenin nuclear translocation. However, IHC analysis has shown that nuclear accumulation of β-catenin is exclusively observed at the invasive front where tumor cells are exposed to growth factors and cytokines secreted by the stromal microenvironment likely to further enhance Wnt signaling in a localized fashion (*Fodde and Brabletz, 2007*). As different levels of Wnt/β-catenin signaling are associated with distinct cellular outcomes (*Gaspar and Fodde, 2004*),

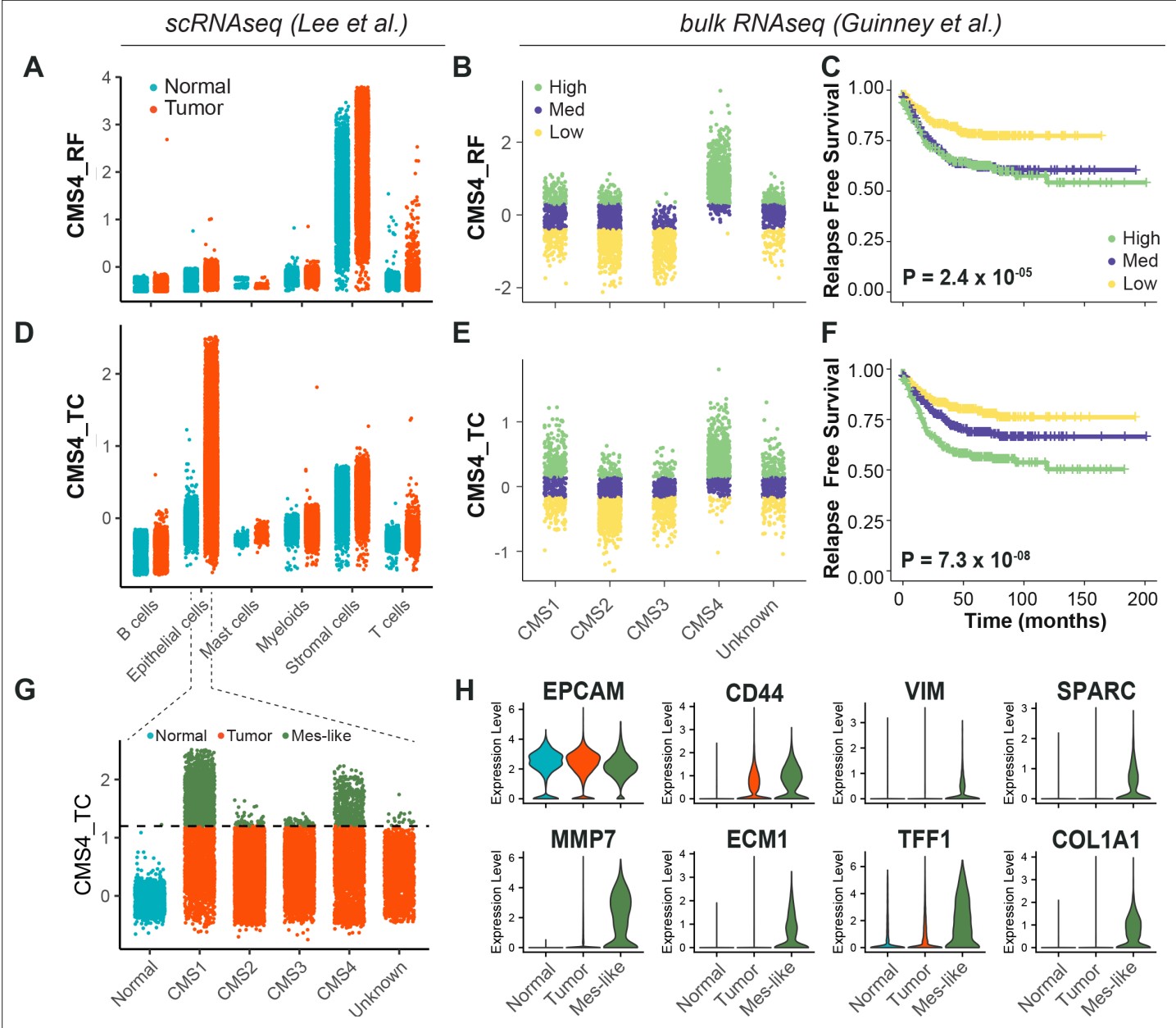

**Figure 6.** Identification of EpCAM$^{lo}$ cells in primary colorectal tumors. (**A**) Expression of the CMS4_RF signature in the scRNAseq data from Lee et al. (N = 91,103 cells) indicates high association to the stromal cells. (**B**) Expression of the CMS4_RF signature in the bulk RNAseq data from Guinney et al. (N = 3232 tumors) shows association to CMS4 tumors. Tumors were grouped in three equal groups according to their association with the CMS4_RF signature. (**C**) Kaplan–Meier plot of the three CMS4_RF groups shows significant differences in relapse-free survival. (**D**) Expression of the CMS4_TC signature in the scRNAseq data reveals high association to the tumor epithelial cells. (**E**) Expression of the CMS4_TC signature in the bulk RNAseq data shows association to CMS1 and CMS4 tumors. Tumors were grouped in three equal groups according to their association with the CMS4_TC signature. (**F**) Kaplan–Meier plot of the three CMS4_TC groups shows significant differences in relapse-free survival. (**G**) Expression of the CMS4_TC signature in the fraction of epithelial cells (N = 24,707 cells). Cells from CMS1 and CMS4 tumors show the highest association to the CMS4_TC signature and were annotated as mes-like. (**H**) Violin plots of normal, tumor, and mes-like tumor cells showing expression patterns across different genes.

The online version of this article includes the following figure supplement(s) for figure 6:

**Figure supplement 1.** Survival analysis using the CMS4_RF and CMS4_TC signatures across the different consensus molecular subtypes.

it is plausible to think that the 'full-blown' activation of this pathway, earmarked by nuclear β-catenin, may trigger EMT and endow phenotypic plasticity in a minority of colon cancer cells located at the

invasive front. The transient and reversible nature of these cellular modifications is likely driven by epigenetic changes at EMT-TFs.

Here, we have taken advantage of the coexistence of biologically distinct cellular phenotypes within commonly employed colon cancer cell lines to study the cellular and molecular mechanisms that underlie phenotypic plasticity and the acquisition of invasive and metastatic capacity. While the bulk of the cancer cells have a characteristic epithelial phenotype (here referred to as EpCAM^hi), a minority of cells with mesenchymal morphology and plastic features (EpCAM^lo) coexist in a dynamic equilibrium through stochastic state transitions with their more committed counterpart. Similar observations have been previously made in breast cancer and oral squamous cell carcinoma cell lines (*Gupta et al., 2011*; *Biddle et al., 2011*), which underlines their relevance for the study of phenotypic plasticity in a broad spectrum of malignancies.

As predicted by their morphology, EpCAM^lo cells feature increased invasive and metastatic capacity and a distinct gene expression profile when compared to their epithelial counterpart. In particular, EMT activation resulting from enhanced Wnt signaling and *ZEB1* upregulation are distinctive features of the EpCAM^lo transcriptome. Of note, *ZEB1* plays many pleiotropic roles ranging from the modulation of oncogenic and tumor-suppressive pathways, cell-fate determination, stemness, and cell plasticity (*Caramel et al., 2017*) and is as such likely to be a determinant in colon cancer invasion and metastasis. From this perspective, the recent debate on EMT as an essential requirement for metastasis (*Maheswaran and Haber, 2015*) likely mirrors the complexity of transcription factors and their downstream targets involved in these processes. Moreover, a recent scRNAseq comparative study of various time-course EMT models has revealed very limited overlap among differentially expressed genes indicative of the vastly context-dependent nature of these processes (*Cook and Vanderhyden, 2020*). The here presented data emphasize the key role played by EMT and its reverse MET in colon tumor cell dissemination and distant metastasis formation.

The observed role of enhanced Wnt signaling in EpCAM^lo cells is of interest in view of the presence of *APC* and *CTNNB1* (β-catenin) gene mutations in these cell lines (*Nishisho et al., 1991*; *Ilyas et al., 1997*). Further enhancement of Wnt signaling is likely to result in EMT activation through *ZEB1* upregulation and its conversion from a repressor to activator state (*Sánchez-Tilló et al., 2011*). On its turn, increased *ZEB1* expression was also shown to enhance Wnt, possibly in an autocrine stimulatory circuit (*Sánchez-Tilló et al., 2015*). Accordingly, Wnt 'super-activation' by GSK3β inhibition (i.e., Chiron) in the *APC/CTNNB1*-mutated colon cancer cell lines results in a significant expansion of EpCAM^lo cells and increased *ZEB1* expression. Vice versa, *ZEB1* downregulation by shRNA leads to a dramatic contraction of the EpCAM^lo subpopulation. Notably, isogenic subclones of HCT116 carrying wild type, heterozygous, and homozygous mutations at the β-catenin locus (*Kim et al., 2019*) show a progressive increase in the size of the EpCAM^lo subpopulation. These observations highlight the relevance of Wnt dosages in the downstream activation of EMT.

Transcriptional regulation of EMT-TFs has been shown to be controlled by miRNAs binding to specific 3′UTR sequences. In particular, members of the miR-200 family inhibit EMT and cancer cell motility by suppressing Wnt/β-catenin signaling and directly interfering with the expression of EMT-TFs and in particular of *ZEB1/2* (*Brabletz and Brabletz, 2010*; *Ghahhari and Babashah, 2015*). Accordingly, we have shown that the expression of different members of the miR-200 family is downregulated in EpCAM^lo colon cancer cells when compared with their epithelial counterpart. Moreover, the RKO cell line, composed entirely of EpCAM^lo cells, was previously reported to be characterized by epigenetic downregulation of miR-200a/b/c, leading to high *ZEB1* expression levels (*Davalos et al., 2012*) in the absence of Wnt-activating mutations (*Sparks et al., 1998*). As the epigenetic activation/silencing of the miR-200 locus is dynamically regulated by common EMT-inducers such as TGFβ and Wnt (*Ghahhari and Babashah, 2015*; *Davalos et al., 2012*), it is likely that phenotypic plasticity is triggered in vivo in cells located at the invasive front of colon carcinomas as a result of a complex interplay between their specific mutation spectra and epigenetic profiles modulated by different signaling cues from the microenvironment. The consistent presence of cells with nuclear co-localization of ZEB1 and β-catenin located at the invasive front of patient-derived colon carcinomas validates these observations. Other EMT-TFs like ZEB2 are likely to be upregulated in ZEB1-negative cases (*Kahlert et al., 2011*).

The clinical relevance of our results, originated from the analysis of immortalized cancer cell lines, came from the bioinformatic mining of bulk and scRNAseq data from patient-derived colon cancers.

First, EpCAM$^{lo}$ expression signatures are highly correlated with the CMS4 signature (**Guinney et al., 2015**) and with shorter disease-free and overall survival. The mesenchymal CMS4 subtype accounts for approximately 25% of the cases and is associated with the greatest propensity to form metastases. While the CMS4 signature was initially attributed to the presence of tumor-associated stroma and not to a cancer cell-intrinsic property (**Calon et al., 2015**; **Isella et al., 2015**), it has been shown that *bona fide* tumor cells from stroma-rich cancers have a markedly mesenchymal gene expression pattern (**Vellinga et al., 2016**). By taking advantage of the Lee et al. scRNAseq study (**Lee et al., 2020**), we derived a tumor-specific signature (*CMS4_TC*) by selecting epithelial-specific genes from the original CMS4_RF signature that outperforms other commonly employed prognostic and predictive markers (e.g., *BRAF* mutation and MSI; *data not shown*) in stratifying CMS1, 3, and 4 colon cancer patients based on overall survival. These results highlight the relevance of our study in the identification of a quasi-mesenchymal cellular state with plastic, invasive, and metastatic properties predictive of poor prognosis in colon cancer patients regardless of their CMS classification.

Recent studies on the role of EMT in eliciting phenotypic plasticity in cancer cells have highlighted the relevance of intermediate cellular stages co-expressing both epithelial and mesenchymal genes for tumor progression and metastasis (**Jolly et al., 2015**; **Aiello et al., 2018**; **Pastushenko et al., 2018**). These hybrid E/M or partial EMT cells are thought to be endowed with increased invasive and metastatic capacity. Our scRNAseq analysis of EpCAM$^{lo}$ colon cancer cells has revealed not only fully mesenchymal but also hybrid E/M subclusters, the latter predicted in silico to underlie the observed transcriptional heterogeneity. Of note, transcriptional activation of specific arrays of E- (epi1/2) and M- (mes1/2) genes accompanies the transition between cellular states. Partial EMT cells, predicted by RNAvelocity to transit between states, are characterized not only by the co-expression of E- and M-specific genes at intermediate levels, but also by increased expression of specific genes like *SPARC* (Secreted Protein Acidic and Rich in Cysteine, also known as osteonectin), encoding for a matricellular protein involved in the modulation of cell–cell and cell–matrix interactions and known as a prognostic marker in colon cancer (**Kim et al., 2013**). The role of SPARC in cancer is controversial as it has been shown to promote EMT and metastasis, but also to encompass tumor-suppressive functions in a context-dependent fashion (**Podhajcer et al., 2008**). Importantly, SPARC triggers EMT through direct cell-to-cell contact and upregulation of other EMT-inducers like fibronectin (FN1) (**Takigawa et al., 2017**), reminiscent of the interactions occurring between parenchymal and stromal cells at the invasive front in colon cancer (**Fodde and Brabletz, 2007**) where pEMT is expected to underlie plasticity and invasion through the ECM. Although SPARC is unique in its expression peaking at pEMT states and earmarking the transition between EpCAM$^{lo}$ and EpCAM$^{hi}$ in both cell lines and in mes-like cells from patient-derived colon cancers, it seems unlikely that specific genes exist that can independently elicit pEMT. Instead, tumor-specific and context-dependent activation of subset of genes with distinct functions (e.g., *SPARC*, *FN1*, *MMP7*, *ZEB1*, *ECM1*) synergistically promoting, for example, collective cell migration upon interaction with the stromal microenvironment, may represent a more likely scenario.

The metastable and plastic features of EpCAM$^{lo}$ cells were further highlighted by their striking capacity of giving rise to distant metastases reminiscent of the primary tumors both in the prevalence of EpCAM$^{hi}$ cells and in the distinct patterns of β-catenin intracellular localization between the periphery and center of the lesions. These observations are in agreement with the key role of MET for the onset of distant metastasis (**Brabletz et al., 2005**). The admittedly marginal increase in liver and lung metastases upon continuous induction of *ZEB1* expression is likely to result from MET inhibition. From this perspective, our results provide support to the 'migrating CSC' (**Brabletz et al., 2005**) and 'β-catenin paradox' (**Fodde and Brabletz, 2007**) models and shed light on the cellular and molecular mechanisms underlying Wnt super-activation, (p)EMT induction, and cell dissemination along the invasive front of colon carcinomas.

The characterization of the cellular and molecular mechanisms underlying phenotypic plasticity in colon cancer cells along the invasion-metastasis cascade has great relevance for the development of future diagnostic and therapeutic approaches based on circulating tumor cells (CTCs). Although current detection methodologies mainly rely on their epithelial features, more recent studies have indicated that CTCs with less pronounced epithelial characteristics are likely to underlie metastasis at distant organ sites. In breast and oral cancer, hybrid E/M tumor cells feature the highest degree of phenotypic plasticity coupling the capacity of undergoing EMT/MET with therapy resistance and

other stem cells features (*Yu et al., 2013*; *Biddle et al., 2016*). Further single-cell RNAseq and functional analyses of extensive cohorts of EpCAM[lo] cells from matched primary tumors, CTCs, and the corresponding distant metastases will open new avenues for diagnostic and therapeutic applications.

## Materials and methods

**Key resources table**

| Reagent type (species) or resource | Designation | Source or reference | Identifiers | Additional information |
|---|---|---|---|---|
| Cell line (*Homo sapiens*) | HCT116 (adult colorectal carcinoma) | ECACC | Cat# 91091005, RRID:CVCL_0291 | For sorting of subpopulations; see Materials and methods, section Flow cytometry and sorting |
| Cell line (*Homo sapiens*) | HCT116-WT, -P, and -MT (wild type, hetero- and homozygous for the b-catenin Ser45del mutant allele) | Author:A.K. (*Kim et al., 2019*) | | |
| Cell line (*Homo sapiens*) | SW480 (adult colorectal carcinoma) | ECACC | Cat# 87092801, RRID:CVCL_0546 | For sorting of subpopulations, see Materials and methods, section Flow cytometry and sorting |
| Biological sample (*Mus musculus*) | APC-Kras-P53 (AKP) organoids (*Apc$^{fl/fl}$::Kras$^{G12D/+}$::Trp53$^{fl/R172H}$*) | Author: J.vR (*Fumagalli et al., 2017*; *Fumagalli et al., 2018*) | | |
| Biological sample(*Mus musculus*) | AKP-Zeb1 (AKP-Z) (dox-inducible *Zeb1* expression) | This study | | See Materials and methods, section Construction of Zeb1 inducible vector |
| Antibody | Anti-mouse CD44-APC (rat monoclonal) | BD Pharmingen | Cat# 559250, RRID:AB_398661 | FACS (1 µg/10$^6$ cells) |
| Antibody | Anti-human EpCAM-FITC (mouse monoclonal) | GeneTex | Cat# GTX30708, RRID:AB_1240769 | FACS (1 µg/10$^6$ cells) |
| Antibody | Anti-human EpCAM (mouse monoclonal) | Santa Cruz Biotechnology | Cat# sc-66020, RRID:AB_2098654 | IF (1:250) |
| Antibody | Anti-human ZEB1 (rabbit polyclonal) | Santa Cruz Biotechnology | Cat# sc-25388, RRID:AB_2217979 | IF (1:200) |
| Antibody | Anti-human β-catenin (mouse monoclonal) | BD Biosciences | Cat# 610154, RRID:AB_397555 | IHC (1:500) |
| Antibody | Anti-human ZEB1 (rabbit polyclonal) | Sigma-Aldrich | Cat# HPA027524, RRID:AB_1844977 | IHC (1:200) |
| Chemical compound, drug | 5-fluorouacil | Sigma-Aldrich | Cat# F6627 | 1.5–100 µg/mL |
| Chemical compound, drug | Oxaliplatin | Sigma-Aldrich | Cat# O9512 | 1.25–20 µg/mL |
| Commercial assay or kit | TruSeq Sample Preparation | Illumina | Cat# 15026495F | v.2 |
| Commercial assay or kit | Single Cell 3′ Reagent Kit Protocol | 10XGenomics | Cat# CG00052 | v2 chemistry |
| Software, algorithm | CellRanger | 10XGenomics | RRID:SCR_017344 | Version 2.1.1 |
| Software, algorithm | R | Seurat, GSVA, MAGIC (*Stuart et al., 2019*; *van Dijk et al., 2018*; *Hanzelmann et al., 2013*) | RRID:SCR_007322, RRID:SCR_021058 | Version 4.0.4 |
| Software, algorithm | Python | Velocyto, scVelo (*La Manno et al., 2018*; *Bergen et al., 2020*) | RRID:SCR_018167, RRID:SCR_018168 | Version 3.8.3 |

## Cell cultures

Human colon cancer cell lines were acquired from the European Collection of Authenticated Cell Cultures (ECACC) and cultured in DMEM medium (11965092, Thermo Fisher Scientific) (supplemented with 10% heat-inactivated fetal bovine serum [FBS; Thermo Fisher Scientific], 1% penicillin/ streptomycin [penicillin: 100 U/mL, streptomycin: 100 µg/mL; 15140122 Thermo Fisher Scientific]) in humidified atmosphere at 37°C and 5% $CO_2$. The identity of each cell line was confirmed by DNA fingerprinting with microsatellite markers (Amelogenin, CSF1PO, D13S317, D16S539, D5S818, D7S820, THO1, TPOX, vWA, D8S1179, FGA, Penta E, Penta D, D18S51, D3S1358, D21S11) and compared with the analogous data provided by ATCC, EACC, and https://web.expasy.org/cello-saurus/ (data not shown). The HCT116-P, HCT116-WT, and HCT116-MT cell lines were kindly given by the laboratory of Hoguen Kim from the Yonsei University College of Medicine in Seoul, Korea. All employed cell lines tested negative for *Mycoplasma*.

$Apc^{fl/fl}::Kras^{G12D/+}::Trp53^{fl/R172H}$ (AKP) organoids were grown in 50 µL droplets of Matrigel (Corning) covered with Advanced DMEM-F12 medium (12634028, Thermo Fisher Scientific) supplemented with 1× GlutaMAX (35050-061, Thermo Fisher Scientific), HEPES (15630-056, Thermo Fisher Scientific), gentamicin (15750060, Thermo Fisher Scientific), 100 ng/mL mEgf (PMG8041, Thermo Fisher Scientific), 2% B27 supplement (12587010, Thermo Fisher Scientific), mNoggin (self-produced according to protocol), and 100 µM Y-27632 (Y0503, Sigma Aldrich).

Chiron (CHIR99021, 4423, Tocris, dissolved at 20 mM in DMSO), 4 µM CHIR99021 was added to the culture medium every 48 hr for 1 week.

## Flow cytometry and sorting

For flow cytometry and sorting purposes, cells were detached from the culture dishes using trypsin-EDTA (15400054, Thermo Fisher Scientific) and resuspended in staining buffer (PBS supplemented with 4% FCS). Antibody staining was performed on ice for 30″ with CD44-APC (clone IM7, 559250, BD Pharmingen) and EpCAM-FITC (ESA-214, GTX30708, GeneTex). Cells were then washed and resuspended in PBS 4% FCS. Flow cytometry analysis and cell sorting were carried out with a FACSAria III Cell Sorter (BD Biosciences).FSC-H versus FSC-W and SSC-H versus SSC-W were employed to eliminate cell aggregates and ensure single-cell sorting. Gates were defined as depicted and specified in *Figure 1*. The DAPI nuclear dye (D9542, Sigma-Aldrich) was used at 1 µg/mL to exclude dead cells. FITC and GFP were analyzed using 488 nm laser and 502 LP and 530/30 BP filters; APC and Sytox Red with a 633 nm laser and a 660/20 BP filter; BV421 using a 405 nm laser and a 450/40 BP filter; BV785 with a 405 nm laser and a 750 LP and 780/60 BP filter; PE with a 461 nm laser and a 582/15 BP filter.

For the clonogenicity assays, single cells were sorted into 96-well plates at 1 cell/well in culture medium. After sorting, the plates were checked under the microscope to confirm the presence of a single cell per well. Upon confluency, each single clone was FACS-analyzed for CD44/EpCAM at different time points.

For the flow cytometry analysis of mouse liver metastases and primary AKP-Z organoid-derived tumors, freshly resected tissue samples were cut into small pieces, washed three times with ice-cold PBS, and subsequently digested in Advanced DMEM-F12 (12634028, Thermo Fisher Scientific) containing 50 mg/mL of collagenase A (11088793001, Roche) for 30′ at 37°C, with vigorous pipetting every 10′. Residual tissue fragments were further treated with TrypLE Express (12605-010, Thermo Fisher Scientific), 0,25% Trypsin (Thermo Fisher Scientific), and 0.1 mg/mL DNaseI (DN25, Sigma-Aldrich) for 45′ at 37°C. Samples were then washed and filtered in sequence through a 100 and 40 µm cell strainer (Corning). The HCT116-derived cells from the liver metastasis were identified and analyzed using the following antibodies for 30′ on ice: HLA-A, B, C-biotin (clone W6/32, 311434, Biolegend), Streptavidin-PE-Dazzle 549 (Biolegend), rat anti-mouse CD45-BV421 (clone 30-F11, 563890, BD Biosciences), rat anti-mouse CD31-BV421 (clone 390, 563356, BD Biosciences), rat anti-mouse TER-199-BV421 (clone TER-119, 563998, BD Biosciences), CD44-APC (clone IM7, 559250, BD Pharmingen) ,and EpCAM-FITC (ESA-214, GTX30708, GeneTex). AKP-Z tumor-derived cells were stained using the following antibodies: rat anti-mouse CD45-BV421 (clone 30-F11, 563890, BD Biosciences), rat anti-mouse CD31-BV421 (clone 390, 563356, BD Biosciences), rat anti-mouse TER-199-BV421 (clone TER-119, 563998, BD Biosciences), anti-mouse EpCAM-PE (clone G8.8, Biolegend), anti-mouse/human

CD44-APC (clone!M7, BD Biosciences), and DAPI. Intrinsic GFP expression was also used to select for organoid-derived cells.

## Cell proliferation assay and cell cycle analysis

For proliferation assays, sorted cells were seeded in 24-well dishes (2500 cells/well) and cultured in conventional medium. Cells were harvested and counted at the indicated time points. For cell cycle analysis, sorted cells were centrifuged, fixed in 70% ethanol, and stained with 50 µg/mL propidium iodide (PI) (Sigma) and 0.2 mg/mL RNAse A (10109142001, Sigma-Aldrich). The stained cells were then FACS-analyzed using a 655 LP and a 695/40 BP filter.

## Cell viability assays

For the chemoresistance assays, cells were seeded in 96-well plates at 15,000 cells/well and left overnight to adhere. Three technical replicates were plated per tested condition. Both oxaliplatin (Sigma-Aldrich) and 5-fluorouacil (5-FU; Sigma-Aldrich) were dissolved in DMSO. HCT116 cells were incubated for 3 or 5 days with 5-FU and oxaliplatin, respectively, whereas SW480 cells were treated with 5-FU or oxaliplatin for 7 days (calibrated on the response of the parental line in terms of viability). After removal of the chemotherapeutic drug, cells were washed with PBS and left to regrow in standard culture medium. Cell viability was assessed using the MTT (3-(4,5-dimethylthiazol-2-yl)–2,5-diphenyltetrazolium bromide; Sigma-Aldrich) assay at time 0, that is, upon removal of the drug, and after regrowth (range 1–36 days). Briefly, cells were incubated at 37°C, 5% $CO_2$ for 3 hr in culture medium supplemented with 0.45 mg/mL MTT. The 96-well plates were then centrifuged at 1000 rpm for 5′ and the culture medium removed. MTT formazan precipitates were solubilized with DMSO. O.D. reading was performed at 595 nm with a Microplate Reader (Model 550, Bio-Rad). Background measurements were subtracted from each data point. MTT-based results were also validated by manual count of trypan blue-positive cells using cells harvested from additional 96 wells. At least two biological experiments were performed for each individual cell line and drug.

Selection of oxaliplatin and 5-FU concentrations and administration times for chemo-resistance assays was as follows. In a predefined range of concentrations (1.5–100 µg/mL for 5-FU and 0.6–50 µg/mL for oxaliplatin) based on previous data, the maximal concentration and length of treatment were selected as the combination able to induce cell death in approximately 80% of EpCAM$^{lo}$ cells. From that concentration, the curve was scaled down by 50% at least three times to reach the lowest concentration to be tested. The minimal treatment length was 72 hr, while the maximal duration was 1 week. Oxaliplatin was considerably slower than 5-FU in its cytotoxicity with HCT116 cells. The difference was less relevant in SW480 cells where numerous cells, although irreversibly damaged, remained viable for several days. For the latter reason, treatment was longer in SW480 cells.

## Migration and invasion assays

For the 'transwell' (modified Boyden chamber) migration and invasion assays, cells were starved for 24 hr in DMEM supplemented with 1% FBS. For each tested subpopulation, $1.0 \times 10^6$ cells were harvested, resuspended in 1% FBS medium, and plated in transwell tissue culture inserts (8 µm polycarbonate membrane, 24 well, 3428; Corning). For the invasion assays, 1.5 µg/mL Matrigel dissolved in serum-free medium was pipetted on top of the transwell membrane and left to set at 37°C for several hours before plating the cells. DMEM medium with 10% FBS was then placed in the lower compartment of the transwell chamber. For both assays, three technical replicates were plated for each subpopulation. Plates were incubated for 24 hr at 37°C/5% $CO_2$, after which the cells and membranes were fixed in methanol and stained with crystal violet. The non-migrated cells on the upper surface of the membrane were removed with a cotton-wool bud and the cells migrated to the lower surface of the membrane counted. For both assays, two independent experiments were performed.

## RNA isolation and qRT-PCR

RNA was isolated from cells using TRIzol Reagent (15596018, Thermo Fisher Scientific) according to the manufacturer's instructions. RNA concentration was measured by NanoDrop. Subsequently, reverse transcription into cDNA was conducted using the High-Capacity cDNA Reverse Transcription Kit (4368814, Life Technologies) according to the manufacturer's instructions. RT-qPCR was performed with Fast SYBR Green Master Mix (4385617; Thermo Fisher Scientific). PCR amplification and detection

were implemented with the 7400 Fast Real-Time PCR System. Expression levels were normalized for the endogenous *GAPDH* reference gene. All PCR primers here employed are listed below:

| Gene | Forwardprimer | Reverseprimer |
|---|---|---|
| *GAPDH* | 5'-ACCCAGAAGACTGTGGATGG-3' | 5'-TCTAGACGGCAGGTCAGGTC-3' |
| *EPCAM* | 5'-GCAGCTCAGGAAGAATGTG-3' | 5'-CAGCCAGCTTTGAGCAAATGAC-3' |
| *CDH1* | 5'-TGCCCAGAAAATGAAAAAGG-3' | 5'-GTGTATGTGGCAATGCGTTC-3' |
| *VIM* | 5'-GAGAACTTTGCCGTTGAAGC-3' | 5'-GCTTCCTGTAGGTGGCAATC-3' |
| *CDH2* | 5'-CAACTTGCCAGAAAACTCCAGG-3' | 5'-ATGAAACCGGGCTATCTGCTC-3' |
| *FN1* | 5'-CAGTGGGAGACCTCGAGAAG-3' | 5'-TCCCTCGGAACATCAGAAAC-3' |
| *ZEB1* | 5'-GCACAACCAAGTGCAGAAGA-3' | 5'-CATTTGCAGATTGAGGCTGA-3' |
| *ZEB2* | 5'-TTCCTGGGCTACGACCATAC-3' | 5'-TGTGCTCCATCAAGCAATTC-3' |
| *TWIST1* | 5'-GTCCGCAGTCTTACGAGGAG-3' | 5'-GCTTGAGGGTCTGAATCTTGCT-3' |
| *SNAI1* | 5'-GCGAGCTGCAGGACTCTAAT-3' | 5'-CCACTGTCCTCATCTGACA-3' |
| *SNAI2* | 5'-GGGGAGAAGCCTTTTTCTTG-3' | 5'-TCCTCATGTTTGTGCAGGAG-3' |
| *FOXC2* | 5'-GCCTAAGGACCTGGTGAAGC-3' | 5'-TTGACGAAGCACTCGTTGAG-3' |

## miRNA

For miRNA experiments, sorted cell populations from HCT116 and SW480 cell lines were cultured up till 70% confluency in 6-well-multiwell plates. Total RNA was then isolated using the Trizol (Ambion) protocol. For *miR-200-family (miR-200a, miR-200b, miR-200c, miR-141, miR-429)*, reverse transcription was performed using the TaqMan MicroRNA RT Kit (Applied Biosystems), followed by qRT-PCR using TaqMan MicroRNA assay (Thermo Fisher Scientific). *U6* snRNA was used as tanendogenous control. The expression of *miR-200-family* was analyzed by the ΔCT method. The experiment was repeated three times.

## shRNA

In order to knock down *ZEB1* expression, lentiviral-inducible shRNA vectors encompassing control or *ZEB1* sequences were developed. Cloning was performed according to the manufacturer's instructions (**Wiederschain et al., 2009**). Briefly, Tet-pLKO-puro (gift from D. Wiederschain, Addgene plasmid # 21915) was digested with AgeI and EcoRI and isolated by gel purification (QIAEX II Gel Extraction Kit, Qiagen). The sequences of the control and *ZEB1* shRNA were as follows: shRNA control (shCT) (Addgene sequence #1864): top 5'-CCGGCCTAAGGTTAAGTCGCCCTCGCTCGAGCGAGGGCGA CTTAACCTTAGGTTTTTTG-3', bottom: 5'-AATTCAAAAAACCTAAGGTTAAGTCGCCCTCGCTCG AGCGAGGGCGACTTAACCTTAGG-3', shZEB1-A: target sequence (Broad Institute): 5'-GCTGCCAA TAAGCAAACGATT-3' oligo sequence: top: 5'-CCGGGCTGCCAATAAGCAAACGATTCTCGAGA ATCGTTTGCTTATTGGCAGCTTTTT-3', bottom: 5'-AATTAAAAAGCTGCCAATAAGCAAACGATTCT CGAGAATCGTTTGCTTATTGGCAGC-3', shZEB1-B: target sequence (Broad Institute): 5'-GTCTGGGT GTAATCGTAAATT-3' oligo sequence: top 5'-CCGGGTCTGGGTGTAATCGTAAATTCTCGAGAATTT ACGATTACACCCAGACTTTTT-3', bottom 5'- AATTAAAAAGTCTGGGTGTAATCGTAAATTCTCGAG AATTTACGATTACACCCAGAC-3'.

The 'top' and 'bottom' labels indicate the oligonucleotide that were annealed in 0.1 M NaCl, 10 mM Tris HCl, pH 7.4, after incubation at 95°C followed by a cooling down step until room temperature is reached. The digested vector (200 ng) was ligated with 1 μL of the oligonucleotides (0.45 nmol/μL) using T4 DNA ligase (Promega) for 3 hr at room temperature. 'One Shot Stbl3' chemically competent *Escherichia coli* (Thermo Fisher Scientific) were transformed with the ligation product. Individual colonies were inoculated for mini-prep cultures, and the presence of the insert was checked using the restriction enzyme XhoI and by sequencing.

For lentivirus production, the shCT or the pool of the two *ZEB1* shRNA constructs were packaged into second-generation virus particles using psPAX2 (Addgene plasmid # 12260; gift from Dr. Didier Trono) and pMD2.G (Addgene plasmid # 12259) into HEK293T. Virus particles were titrated with

the HCT116 cell line and a MOI of 0.5 was employed to produce the shRNA-inducible HCT116 and SW480 cell lines. Lentivirus-infected cells were selected in medium containing 1 μg/mL puromycin (Dulbecco). shRNA induction was implemented by using 1 μg/mL doxycycline for 72 hr. The extent of *ZEB1* downregulation was assessed by RT-qPCR.

## Construction of Zeb1-inducible vector

A pORF mZeb1 plasmid (Cat no. ORF062179) encompassing the cDNA of the *Zeb1* gene was employed for the construction of the lentiviral vector. The *Zeb1* gene was cloned into the pEN_ TmiRc3 plasmid, gift from Iain Fraser (California Institute of Technology, CA). The *Zeb1* insert was transferred into a pSLIK-Hygro plasmid (#25737; Addgene, USA) by Gateway technology. All newly generated plasmids were sequence-verified by LGC Genomics (LGC Genomics GmbH, Germany). The pSLIK-Hygro plasmid was transiently transfected by Fugene HD (Promega) in HEK293T cells together with the VSV-G, MD, and REV packaging plasmids. After 2 days, the culture medium was collected and viral particles were concentrated by ultracentrifugation. AKP organoids were transduced with the concentrated virus. After 24 hr, the transduced cells were selected by hygromycin B (Thermo Fisher Scientific, The Netherlands) (60 μg/mL) for 7 days. Expression of *Zeb1* was confirmed by qRT–PCR upon stimulation with doxycycline hyclate.

## Animal experiments

All protocols involving animals were approved by the Dutch Animal Experimental Committee and conformed to the Code of Practice for Animal Experiments in Cancer Research established by the Netherlands Inspectorate for Health Protections, Commodities and Veterinary Public health (The Hague, the Netherlands, 1999). Animals were bred and maintained in the Erasmus MC animal facility (EDC) under conventional specific pathogen-free (SPF) conditions.

Spleen transplantation assays were implemented on 6- to 8-week-old NOD.Cg-*Prkdc^{scid} Il2rg^{tm1Wjl}*/SzJ (NSG) male and female mice anesthetized intraperitoneally with ketamine (Ketalin, 0.12 mg/mL) and xylazine (Rompun, 0.61 mg/mL). Carpofen (Rimadyl, 5 mg/mL) was given subcutaneously as analgesia. The spleen was exteriorized through a left lateral flank incision and $2.5 \times 10^4$ HCT116 and SW480 cells, resuspended in 50 μL of PBS, were injected into the spleen parenchyma using an insulin syringe. Fifteen minutes after injection, a splenectomy was performed with a fine tip cautery pen in order to remove spilled cells and ensure hemostasis. The peritoneum and skin were then sutured in two layers. Mice injected with HCT116 cells were sacrificed 4 weeks after injection for tumor collection; mice injected with SW480 cells were killed 8 weeks after spleen transplantation. Upon liver resection, individual macroscopic metastatic lesions were counted, dissected, and fixed in 4% paraformaldehyde (4% PFA). Residual liver tissue and the lungs were also fixed in 4% PFA for further immunohistochemical analyses.

Orthotopic transplantation of intestinal mouse organoids was performed as previously described (*Fumagalli et al., 2018*). In brief, the day before transplantation AKP-Zeb1 organoids containing a GFP and click beetle luciferase vector (ATG-1929, a gift from Keith Wood, Addgene plasmid #108712) were collected and mechanically dissociated into small clumps. About 100,000 cells were plated in 10 μL drops neutralized Rat Tail High Concentrate Type I Collagen (Corning, Cat. no. 354249) and let to recover overnight at 37°C in Advanced DMEM-F12 medium (12634028, Thermo Fisher Scientific) 100 ng/mL mEgf (PMG8041, Thermo Fisher), 2% B27 supplement (12587010, Thermo Fisher), mNoggin (self-produced according to protocol), and 100 μM Y-27632 (Y0503, Sigma Aldrich). Caecum transplantation assays were implemented on 6- to 8-week-old male NOD.Cg-*Prkdc^{scid} Il2rg^{tm1Wjl}*/SzJ (NSG) mice anesthetized intraperitoneally with ketamine (Ketalin, 0.12 mg/mL) and xylazine (Rompun, 0.61 mg/mL). Carpofen (Rimadyl, 5 mg/mL) was given subcutaneously as analgesia. The caecum was exposed through a midline abdominal incision and a collagen drop containing tumor cells was surgically transplanted in the caecal submucosa. The peritoneum and skin were then sutured in two layers. Tumor growth was monitored by abdominal palpation. Mice were sacrificed 6–8 weeks after transplantation. Upon collection, primary caecal tumors were single-cell digested and further analyzed by FACS. Mice used for quantification of liver and lung metastases were injected with Luciferin-D and imaged with an IVIS Spectrum imaging system (Caliper Life Sciences, Hopkinton, MA, USA). After imaging tissues were fixed and cut into 500 μm slices using a Fibrotome, processed for IHC, stained for β-catenin visualize tumor cells, scanned using a NanoZoomer, and counted using NDP view software.

## Immunofluorescence analysis

Coverslips containing a monolayer of cancer cells were fixed for 30′ in 4% PFA at 37°C and washed twice with PBS. Cells were first permeabilized for 15 min at room temperature with 0.2% of Triton X-100 and then incubated in blocking buffer (5% milk powder in PBS-Tween) for 1 hr at room temperature. Cells were then exposed overnight at 4°C to primary antibodies against EpCAM (mouse, 1:250; sc-66020; Santa Cruz Biotechnology) and ZEB1 (rabbit, 1:200; sc-25388, Santa Cruz Biotechnology). After washing twice with PBS-Tween, coverslips were incubated for 1 hr at room temperature in blocking buffer containing the following secondary antibodies: Goat anti-Rabbit Alexa Fluor 594 conjugate (1:250, #A-11037, Life Technologies) and Donkey anti-Mouse Alexa Fluor 488 conjugate (1:250, #A-21202, Life Technologies). Cells were counterstained with DAPI to visualize the nuclei. Coverslips were mounted in VECTAHIELD HardSet Antifade Mounting Medium (#H-1400, Vector Labs) and imaged with a Zeiss LSM-700 confocal microscope. Images were processed with ImageJ (U.S. National Institutes of Health, Bethesda, MD, USA).

## Immunohistochemistry analysis

Tissues from animal experiments were fixed overnight in 4% PFA and embedded in paraffin. Paraffin blocks containing human colon cancer tissue were obtained from the Department of Pathology at the Erasmus Medical Center in Rotterdam. 4 µm sections were mounted on slides. IHC was performed using the EnVision Plus-HRP system (Dako) and antibodies directed against β-catenin (1:200, 610154, BD Biosciences) and ZEB1 (1:200, HPA027524, Sigma-Aldrich). Briefly, paraffin-embedded sections were dewaxed with Xylene and hydrated in 100 and 70% ethanol. Antigen retrieval was performed using pressure cooker pretreatment in a citrate buffer (pH 6.0) for ZEB1 and in a Tris-EDTA buffer (pH 9.0) for the anti-human-mitochondria and anti-β-catenin antibodies. Subsequently, slides were incubated at room temperature in 3% hydrogen peroxidase for 15′ to block endogenous peroxidase activity. Tissue sections were washed and blocked with 5% BSA in PBS-Tween for 1 hr to then be incubated with the primary antibodies overnight at 4°C. Slides were washed twice with PBS-Tween and incubated with Rabbit EnVision+ System HRP (K4001, Dako) or Mouse EnVision+ System HRP (K4007, Dako) for 30′. Subsequently, signal detection was done and tissues were counterstained with Mayer's Hematoxylin. Dehydration was performed by incubation in 70 and 100% ethanol followed by Xylene before sleds were mounted using Pertex (00811, Histolab).

For the IHC analysis of patient-derived colon cancers, all paraffin blocks were collected anonymously from the archives of the Department of Pathology of the Erasmus MC.

## TOP-Flash reporter assay

For the β-catenin/TCF reporter assay (TOP-Flash reporter assay), cells were plated on 48-well dishes and cultured in medium with or without 4 µM CHIR99021. After 48 hr, when 70% confluence was reached, cells were transfected by Fugene HD (Promega) with 125 ng of the TOP-Flash or FOP-Flash reporter constructs together with 25 ng of the Renilla luciferase vector for normalization purposes. Luciferase activity was measured using the Dual-Luciferase Reporter Assay System (Promega) 24 hr post-transfection. Luminescence was measured using a GloMax Luminometer.

## Next-generation sequencing (NGS): RNAseq

RNA quality and quantity was evaluated on a 2100 Bio-analyzer (Agilent) using the Agilent RNA 6000 Pico Kit. RNA samples were further processes according to the TruSeq Sample Preparation v.2 Guide (Illumina) and paired end-sequenced on the HiSeq 2,500 (Illumina).

Illumina paired-end reads of 76 bases were trimmed by removing the TrueSeq adapter sequences using Trimmomatic (v.0.33) (*Bolger et al., 2014*). Subsequently, the reads were mapped in a two-pass procedure to the human reference genome build hg38 with the RNA-seq aligner STAR (v2.4.2a) (*Dobin et al., 2013*) and the *Homo sapiens* GENCODE v23 annotation (*Harrow et al., 2012*). Raw counts were summed with the summarize overlaps function with union mode from the Bioconductor Genomic Alignments package (*Gentleman et al., 2004*) (v1.14.0). Genes were called differentially expressed with a generalized linear model using a negative binomial distribution with correcting for cell lines in multi-cell line comparisons. DESeq2 (v1.16.1) was used to perform these calculations (*Love et al., 2014*). We applied a Wald test to identify statistically significant differently expressed genes. p-values were adjusted using the Benjamini–Hochberg (*Benjamini and Hochberg, 1995*) correction

based on which a threshold value was set at <0.01. MDS was performed after the read counts were normalized with blind variance stabilizing log2 transformation function of DESeq2. Gene Ontology (GO) and Kyoto Encyclopedia of Genes and Genomes (KEGG) gene enrichment analyses were carried out as described previously (*Meinders et al., 2015*). R (v 3.4.0) (R Core Team, 2017; https://www.R-project.org/) was employed for statistical analysis and visualization of the data.

### Bioinformatics analysis bulk RNAseq

For pathway analysis, generated RNAseq datasets were uploaded into Ingenuity Pathway Analysis software (Qiagen). For other bioinformatics analyses, the generated datasets were uploaded into the R2 Genomics Analysis and Visualization Platform (http://r2.amc.nl). First, we used the 'differential expression between groups' option to identify the hundred genes with highest expression in the EpCAM$^{lo}$ fraction in both the HCT116 and SW480 cells. These genes were saved as separate gene sets. Expression values of all genes in both gene sets were then assessed in the CMS3232 composite cohort (*Guinney et al., 2015*), as well as in a large cohort of profiled cell lines originating from the Broad institute. This yielded single meta-gene expression values per tumor or cell line. These gene set expression values were then stored as separate tracks and compared using the 'relate two tracks' option, yielding Pearson r values and accompanying p values.

The 'relate two tracks' option was also used to compare the two gene sets identifying the EpCAM$^{lo}$ cells with gene sets positively identifying the four different molecular subtypes within the published 273-gene CMS classifier, again yielding r values and corresponding p-values.

The gene sets identifying the EpCAM$^{lo}$ cells from both cell lines were also used to cluster the tumors in the CMS3232 cohort into low, intermediate, and high expression groups by k-means clustering. The Kaplan–Meier method was subsequently used to assess significant differences in survival between the generated subgroups. As all tumors in this cohort had previously been assigned to specific CMS subgroups, we then analyzed the contribution of each CMS subtype to each of the generated low, intermediate, and high expression subgroups.

### Single-cell RNAseq

Cell lines were brought to 60–70% confluency before the start of the experiment. For each sample, between $5 \times 10^4$ and $1 \times 10^5$ EpCAM$^{lo}$ and EpCAM$^{hi}$ cells were FACS sorted and processed using the 10 X Genomics Chromium Single Cell Controller. Samples were deep-sequenced (Illumina) to a depth ranging 49–65k reads/cells. Gene-cell matrices were obtained by conversion of the raw data using the Cell Ranger pipeline. Filtered gene-cell matrices were merged in R and processed for downstream analysis using the Seurat package (*Stuart et al., 2019*). Dimension reduction was performed using PCA, tSNE, and UMAP. Epithelial and mesenchymal scores were computed using the Rmagic (imputation) (*van Dijk et al., 2018*) and GSVA (*Hanzelmann et al., 2013*) (scoring) packages. RNA velocity analysis was done in Python using Velocyto (*La Manno et al., 2018*) and scVelo (*Bergen et al., 2020*) packages.

### Statistical analysis

For each experiment, data are shown as mean ± SD. IBM SPSS Statistics software was used for data analysis. The Mann–Whitney U test was used to analyze the difference between two groups of quantitative variables; α-value was set at 5%.

## Acknowledgements

This study has been made possible by funding to RF from the Dutch Digestive Foundation (FP 15-09), Dutch Cancer Society, and the Erasmus MC (EMCR 2015-8090). Also, by funding to MP and OS from the Cancer Research UK core funding to the CRUK Beatson Institure (A17196) and to OS (A21139). The authors are grateful to Drs. R Smits, M Trerotola, S Alberti, and Catherine Winchester for their advice, to Dr. Eric Bindels for assistance with RNAseq analysis, and to 10 X Genomics for their support through their Pilot Award Grant Program.

## Additional information

### Competing interests
Andrea Sacchetti: The authors of this manuscript do not have any competing financial interests in relation to the work described.. Martin M Watson: The other author declares that no competing interests exist.

### Funding

| Funder | Grant reference number | Author |
|---|---|---|
| Maag Lever Darm Stichting | FP 15-09 | Riccardo Fodde |
| KWF Kankerbestrijding | EMCR 2015-8090 | Riccardo Fodde |
| Cancer Research UK | A17196 | Owen J Sansom |
| Cancer Research UK | A21139 | Owen J Sansom |

The funders had no role in study design, data collection and interpretation, or the decision to submit the work for publication.

### Author contributions
Andrea Sacchetti, Miriam Teeuwssen, conceptualization, formal-analysis, Investigation, Methodology, validation, visualization, writing-original-draft; Mathijs Verhagen, conceptualization, formal-analysis, Investigation, software, visualization, writing-original-draft; Rosalie Joosten, Investigation, Methodology, project-administration, validation; Tong Xu, Roberto Stabile, Berdine van der Steen, Martin M Watson, Alem Gusinac, Investigation, Methodology; Won Kyu Kim, Madelon Paauwe, Resources; Inge Ubink, Investigation, Resources; Harmen JG Van de Werken, Investigation, Methodology, Resources; Arianna Fumagalli, Methodology, Resources; Jacco Van Rheenen, Onno Kranenburg, Methodology, Resources, Writing – review and editing; Owen J Sansom, Resources, Writing – review and editing; Riccardo Fodde, conceptualization, data-curation, formal-analysis, funding-acquisition, project-administration, Resources, supervision, visualization, writing-original-draft, Writing – review and editing

### Author ORCIDs
Mathijs Verhagen ⓘ http://orcid.org/0000-0003-3126-8379
Harmen JG Van de Werken ⓘ http://orcid.org/0000-0002-9794-1477
Owen J Sansom ⓘ http://orcid.org/0000-0001-9540-3010
Riccardo Fodde ⓘ http://orcid.org/0000-0001-9839-4324

### Ethics
All protocols involving animals were approved by the Dutch Animal Experimental Committee and were conformed to the Code of Practice for Animal Experiments in Cancer Research established by the Netherlands Inspectorate for Health Protections, Commodities and Veterinary Public health (The Hague, the Netherlands, 1999).

### Decision letter and Author response
Decision letter https://doi.org/10.7554/eLife.61461.sa1
Author response https://doi.org/10.7554/eLife.61461.sa2

## Additional files

### Supplementary files
• Transparent reporting form

### Data availability
The RNA-sequencing data from this study have been submitted to the Gene Expression Omnibus (GEO) database under the accession number GSE154927 and GSE154930 for the bulk and single-cell RNAseq data, respectively.

The following dataset was generated:

| Author(s) | Year | Dataset title | Dataset URL | Database and Identifier |
|---|---|---|---|---|
| Verhagen MP, Teeuwssen MJ, Sacchetti A, van de Werken HG, Fodde R | 2020 | | https://www.ncbi.nlm.nih.gov/geo/query/acc.cgi?acc=GSE154927 | NCBI Gene Expression Omnibus, GSE154927 |
| Verhagen MP, Teeuwssen MJ, Sacchetti A, van de Werken HG, Fodde R | 2020 | | https://www.ncbi.nlm.nih.gov/geo/query/acc.cgi?acc=GSE154930 | NCBI Gene Expression Omnibus, GSE154930 |

The following previously published datasets were used:

| Author(s) | Year | Dataset title | Dataset URL | Database and Identifier |
|---|---|---|---|---|
| Hong Y, Etlioglu HE, Pomella V, van den Bosch B, Vanhecke J, Tejpar S, Boeckx B, Lambrechts D | 2020 | | https://www.ncbi.nlm.nih.gov/geo/query/acc.cgi?acc=GSE144735 | NCBI Gene Expression Omnibus, GSE144735 |
| Hong Y, Lee H, Cho YB, Park W | 2020 | | https://www.ncbi.nlm.nih.gov/geo/query/acc.cgi?acc=GSE132465 | NCBI Gene Expression Omnibus, GSE132465 |
| Guinney J, Dienstmann R, Wang X | 2014 | | https://www.synapse.org/#!Synapse:syn2623706/wiki/67246 | Synapse, 10.7303/syn2623706 |

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
