## [Decision Letter]

**Acceptance summary:**

This paper characterises a sub-population of colon cancer cells that display phenotypic plasticity. These so-called EopCAM^lo^ cells are motile, invasive, chemo-resistant and metastatic. RNA sequencing showed they are associated with enhanced Wnt/b-catenin signalling, plastic (partial) epithelial/mesenchymal state, and subtypes of colon cancer that have poor prognosis. Therefore, distinct sets of epithelial and mesenchymal genes in cells at the invasive front, and in metastasis, define transcriptional trajectories, the epithelial-mesenchymal state, and disease outcome.

**Decision letter after peer review:**

Thank you for submitting your article "Phenotypic plasticity and partial EMT underlie local invasion and distant metastasis in colon cancer" for consideration by *eLife*. Your article has been reviewed by 3 peer reviewers, and the evaluation has been overseen by a Reviewing Editor and Anna Akhmanova as the Senior Editor.

The reviewers have discussed the reviews with one another and the Reviewing Editor has drafted this decision to help you prepare a revised submission.

Summary:

This paper has now been reviewed by three reviewers and myself. While all reviewers see merit in aspects of the work, and indeed the consensus that there were elements of novelty and interest in this manuscript, they felt that novel advances were limited as presented. Briefly, the manuscript falls into two parts; it is too long with too much data presented and we recommend focus on potentially the most exciting/novel part, ie. the RNAseq / sc and computational analyses, and extending this to provide further functional validation. Some of the earlier figures reflect quite well understood biology (EMT, Zeb1, Wnt etc in EMT), and would require much more work to tighten up the conclusions (see Essential revisions below); therefore, it was felt that even if these were improved, the data would likely confirm this confirmed a lot of what we know already. It is true that the role of EMT is controversial – but what is presented in the first part of the manuscript does not add much definitive new data to inform that debate, and indeed the authors' submission letter refers to their 'confirmatory' nature.

We encourage the authors to substantially revise the submission to present a lower number of Figures that focus on the more novel and exciting aspects of the work; perhaps one or two figures defining the cells and their states that they subsequently use for the detailed RNAseq molecular analyses, and then prioritise the molecular and computational analyses, but with a requirement that they take this further and perform some functional validation of molecular signatures, testing whether they have biological meaning as regards defining plasticity or its role – which would be important and novel. The authors should consider whether, and which, earlier figures, and related supplementary figures, could be left out or to streamline and focus the paper. The major points that the authors might consider (below) therefore relate mainly to the definitions of cell sub-populations and enhancing findings and their biological meaning of the scRNA analyses.

Essential revisions:

1. Could the authors introduce plasticity and what they think means in the Introduction. Is reversibility of EMT really an adequate description?

2. Regarding the EpCAM-low cells, why do these get lost during long-term culture and turn into EpCAM-high, E-cadherin-high cells? How then is the homeostasis between the EpCAM-high and -low populations maintained in the parental cells which have been cultured for decades? Also, it seems that almost all single cell cones of EpCAM-low cells turn into EpCAM-high over time. Why are some maintaining the EpCAM-low status? Is there a difference in gene expression or epigenetic imprints?

3. Is there any evidence of EpCAM-low cells in primary tumours in human or mice models? For example, recent work in colon showed that Notch signalling drives colonic to liver metastasis (Jackstadt et al. 2019) – might the Notch-active cells in this model have lower EpCam levels?

4. If the authors choose to retain the metastasis experiments in a revised submission, then they should test the effect of withdrawing Dox after metastatic colonisation to favour proliferation following metastatic seeding? It would be interesting to know whether the timing of Zeb1 expression is important. Does conditional depletion of ZEB1 result in death of these EpCAM-low cells?

5. In the AKP organoids, are the EpCAM-low cells also present and what is the endogenous expression status of ZEB1 in these? Why have the authors maintained the conditional expression of ZEB1 induced in the AKP-Z organoid transplantation experiments? This is likely driving the epithelial cells into a locked mesenchymal state – which is not compatible with the earlier observed 'plasticity' with the EpCAM-low cells in SW480 and HCT116 cells. Since EMT is generally believed to be essential for metastasis formation, the authors should consider temporal ZEB1 expression control in transplanted AKP-Z organoids if these experiments are to be retained in a revised submission.

6. In Figure 7, RNA sequencing identifies that Wnt signaling is likely enhanced in EpCAM-low cells. GSK inhibition induces the expression of ZEB1 (as known before), yet this works only in HCT116 and not in SW480 cells, which actually show an induction of Wnt signaling. The results seem to indicate that there is not just a mere enhancement of Wnt signaling and that other changes/pathways are required as well. Have the authors looked in other cell lines?

7. As Wnt signalling is proposed to be important in the establishment of the EpCam-low population, have the authors inhibited this pathway (either at the ligand level or though inhibiting b-catenin transcription) to confirm that the population is Wnt responsive?

8. The scRNA sequencing seems to reflect EMT full and hybrid stages. The computational analysis is impressive and exciting, the potential trajectories offer a working model which could be experimentally tested by functional validation of the subgroups to finally pinpoint the cell populations with the highest cell plasticity. Most importantly, what defines cell plasticity at the molecular and cellular level? Is it Wnt signaling or something in addition (see point 7 above)? The reader is left without a clear answer. In the scRNA seq, in the populations that have increased EMT and EMT-gene expression, does this correlate to a Wnt/B-catenin signature at a single cell level?

9. What is the mechanistic basis for the "further enhancement" of Wnt signaling? Is it the dose of Wnt signaling or is it the combination with other signaling pathways which cooperate with Wnt transcriptional control, such as Hippo or TGFb signaling? There could be a hint from the RNA sequencing data to distinguish these possibilities. Do the target gene lists change with the enhancement of Wnt signaling?

10. Is the prognostic and predictive value for the gene signature only true for CMS4 CRCs or for all subtypes? Does the EpCAM-low signature and the signatures of the various EMT stages correlate with CMS subtypes, therapy resistance or clinical outcome? This is not really clear from the data presented.

11. The data depicted in Figure 10A and B are confusing and deserve better explanation. How is it possible that EpCAM-low and EpCAM-high sorted cells show overlapping single cell expression profiles upon t-sne plotting in particular for the SW480 cells? This is contradictory to the earlier claims of the authors in the manuscript that EpCAM-low cells have a more mesenchymal gene expression profile. Is there a difference between EpCAM protein expression and EpCAM mRNA expression? Regarding the heatmap from the EMT signature shown in figure 10B, which cell line is represented?

12. The authors link the gene expression signature of EpCAM-low cells with the colon cancer consensus molecular subtype CMS4 which have the worst relapse free and overall survival (Dienstmann R et al. 2017; 17, Nat Rev Cancer 79-92). There are multiple lines of evidence that the mesenchymal signature in CMS4 colon cancers is due to profound infiltration of stromal cells (CAFs, immune cells), extracellular matrix remodeling, TGF-β pathway activation and not the consequence of EMT in cancers cells (e.g. Calon et al. 2015; DOI: 10.1038/ng.3225). It is of course possible that a few epithelial cells in this inflammatory context are undergoing a partial EMT but there is little evidence and this likely will happen in a minority of cells. Therefore, the authors should revise the manuscript manuscript regarding (partial) EMT and the CMS4 and put their findings in a more critical context of what is already in the literature.

---

## [Author Response]

Summary:This paper has now been reviewed by three reviewers and myself. While all reviewers see merit in aspects of the work, and indeed the consensus that there were elements of novelty and interest in this manuscript, they felt that novel advances were limited as presented. Briefly, the manuscript falls into two parts; it is too long with too much data presented and we recommend focus on potentially the most exciting/novel part, ie. the RNAseq / sc and computational analyses, and extending this to provide further functional validation. Some of the earlier figures reflect quite well understood biology (EMT, Zeb1, Wnt etc in EMT), and would require much more work to tighten up the conclusions (see Essential revisions below); therefore, it was felt that even if these were improved, the data would likely confirm this confirmed a lot of what we know already. It is true that the role of EMT is controversial – but what is presented in the first part of the manuscript does not add much definitive new data to inform that debate, and indeed the authors' submission letter refers to their 'confirmatory' nature.We encourage the authors to substantially revise the submission to present a lower number of Figures that focus on the more novel and exciting aspects of the work; perhaps one or two figures defining the cells and their states that they subsequently use for the detailed RNAseq molecular analyses, and then prioritise the molecular and computational analyses, but with a requirement that they take this further and perform some functional validation of molecular signatures, testing whether they have biological meaning as regards defining plasticity or its role – which would be important and novel. The authors should consider whether, and which, earlier figures, and related supplementary figures, could be left out or to streamline and focus the paper. The major points that the authors might consider (below) therefore relate mainly to the definitions of cell sub-populations and enhancing findings and their biological meaning of the scRNA analyses.

We thank the editors for these comments and for the time allocated to the authors for the rebuttal. Although we respectfully disagree with the statements according to which the data reflects on “*well-understood biology*” and “*novel advances were limited as presented*”, the first part of the Results section, i.e. the characterization of the EMT-competent EpCAM^lo^ subpopulation of colon cancer cells, is admittedly of confirmatory nature but nonetheless essential as that these data lay the very basis for the more novel RNAseq-based results. In response to the editors’ requests, we have accordingly shortened this part, drastically reduced the number of main figures (now 6), and moved much of the “incremental” evidence to supplementary figures. The revised manuscript is more focussed on the RNAseq analysis which has now been substantially extended and improved (also based on the reviewers’ comments and suggestions) with some novel and exciting results on partial EMT and phenotypic plasticity, and the role therein of specific genes like *SPARC* (osteonectin), and on the validation of the findings in patient-derived tumors through the generation of a novel classifier based on genes expressed in quasi-mesenchymal tumor cells that outperforms existing ones.

As for the in vivo transplantations, as also commented in reply to the reviewers’ major points, the essential issue that these experiments were meant to address was how *ZEB1* expression affects local invasion and distant metastasis formation in vivo, namely in a model unrelated to the two original cell lines (i.e. the cecum transplantations of AKP-Z mouse organoids). While *ZEB1* expression in the primary tumor does result in a substantial increase of the EpCAM^lo^ subpopulation of quasi-mesenchymal cells, long-term doxycycline administration increases only marginally the multiplicity of lung and liver metastases, likely to result from the *ZEB1*-driven inhibition of MET, i.e. the reacquisition of epithelial features essential for metastatic colonization. To date, our attempts to fine-tune dox administration so to enhance partial EMT and distant metastasis have not succeeded. This is due to the intrinsic variability of the orthotopic transplantation procedure and to the limited number of ‘pulsed’ administrations tested. Additional experiments will require a considerable period of time and very large numbers of mice. Therefore, in agreement with the reviewers’ proposal, we have moved these experiments to the first part of the Results (**Figure S1-5C-E**), and have concentrated on the single cell and bulk expression data, and the development of highly predictive gene signatures as an alternative approach to the clinical validation of our results.

Essential revisions:1. Could the authors introduce plasticity and what they think means in the Introduction. Is reversibility of EMT really an adequate description?

In the revised manuscript, we have defined phenotypic plasticity as the cell’s ability to modify its phenotype (i.e. morphology, physiology, behaviour) in response to cues from the environment in a context-dependent fashion. More than “reversibility of EMT” which is quite restrictive to canonical EMT as we know it, it is the epigenetic (and as such reversible) nature of the global underlying cellular and molecular mechanisms that is quintessential to plasticity. In support of this, a recent scRNAseq comparative analysis of various time-course EMT models has revealed very limited overlap among differentially expressed genes indicative of the vastly context-dependent nature of these processes (*Nat Comm* 2020, 11:2142). To quote from the paper’s abstract: “EMT is not simply a single, linear process, but is highly variable and modular, warranting quantitative frameworks for understanding nuances of the transition.” As such, plasticity may indeed overlap with EMT reversibility, provided its more comprehensive definition is in place.

2. Regarding the EpCAM-low cells, why do these get lost during long-term culture and turn into EpCAM-high, E-cadherin-high cells? How then is the homeostasis between the EpCAM-high and -low populations maintained in the parental cells which have been cultured for decades? Also, it seems that almost all single cell cones of EpCAM-low cells turn into EpCAM-high over time. Why are some maintaining the EpCAM-low status? Is there a difference in gene expression or epigenetic imprints?

Many of the reviewer’s questions were originally addressed by the study by Eric Lander and colleagues (*Cell* 2011, 146:633) reporting for the first time on the existence of distinct subpopulations of cells with stemlike and more epithelial features within human breast cancer cell lines. According to this study, stochastic transitions between states, supported by a Markov model, underlie the phenotypic equilibrium among these subpopulations of cancer cells. The same type of heterogeneity and state transitions seem to apply for the SW480 and HCT116 colon cancer cell lines as presented in our manuscript (**Figure 2). Please note that EpCAM^lo^ cells do not “*get lost*”: just like their EpCAM^hi^ counterpart, they have the ability to modify their defining features and transit to the other identity, albeit with different transition probabilities which ensures that an equilibrium is maintained over time, independently of the initial proportions of the subpopulations. Based on our FACS data, we estimated the average transition probabilities with a two-state Markov model (Figure 2C and S2-1C-D). Notably, the model indicated that EpCAM^lo^ cells have a slightly increased plasticity when compared with their EpCAM^hi^ equivalents (HCT116: 8 fold, SW480: 4 fold). Due to the observed differences in doubling times between the two states, subclones with a lower EpCAMhi>lo transition probability will experience a slight growth advantage which will become prevalent in the long run. We ran a simulation of this effect by starting from a culture with multiple subclones having distinct transition probabilities, indicating that subclones with lower plasticity gain dominance within a few months (Figure S2**). Consequently, especially in the long run, the percentage of EpCAM^lo^ cells will decrease as observed in late cultures.

To elucidate on the plastic potential at the single cell level, we used *RNA velocity* to project the future cellstates. This analysis revealed that cells with partial EMT features has highly plastic capacity as evidenced by the opposing arrows on the RNA velocity plot (**Figure 5E), and that both populations have a trajectory that could “lock” cells in their identity, therefore preventing plastic potential as observed in some of the clones (Figure 2B and S2-1C).**

3. Is there any evidence of EpCAM-low cells in primary tumours in human or mice models?

Yes, the existence of EpCAM^lo^ cells in primary human or murine tumors is shown by several lines of evidence: 1. IHC analysis of ZEB1 (i.e. the main driver of the EpCAM^lo^ cellular identity), β-catenin (whose nuclear localization earmarks enhanced Wnt signalling, a distinctive EpCAM^lo^ feature) revealed how colon cancer cells located at the invasive front express and activate these genes and their corresponding signalling pathway in coordinated fashion. 2. primary mouse tumors (obtained by orthotopic transplantation of genetically modified tumoroids, i.e. independent of the original SW480 and HCT116 cell lines) clearly show a pronounced EpCAM^lo^ subpopulation upon induction of ZEB1 expression (**Figure S1-6C). 3. As now included in the revised manuscript, the epithelial-specific CMS4_TC signature, highly predictive of poor prognosis across the broad spectrum of colon cancers, highlight the existence of *bona fide* quasi-mesenchymal tumor cells that underlie local invasion and distant metastasis (Figure 6 and S6-1).**

Overall, when evaluating the relevance of our results for colon cancer as we see it in patients, one should always keep in mind that the original observation, i.e. the identification of the EpCAM^lo^ cells, was based on two immortalized cell lines where, in the absence of the cues from the tumor microenvironment, plasticity is governed by cell-autonomous processes presumably arisen in vitro. These processes are nonetheless likely to reflect the paracrine interactions that occur between tumor cells and their direct microenvironment. Therefore, to validate their clinical relevance, one must extrapolate the results by looking at the activated pathways and their functional consequences, rather than by focusing on specific combination of genes found in the cell lines.

For example, recent work in colon showed that Notch signalling drives colonic to liver metastasis (Jackstadt et al. 2019) – might the Notch-active cells in this model have lower EpCam levels?

Although this is an interesting question, Notch signalling did not emerge from our RNAseq analyses and as such it does not directly address the issue of the existence of EpCAM^lo^ cells in primary tumors. Nonetheless, together with our main collaborator and co-author Owen Sansom (senior author of the 2019 Jackstadt et al. study), we did mine RNAseq data from different tumor models and organoids mutant in *Kras*, *Apc*, *Braf*, and *Notch*. Notably, a variation in EpCAM expression levels across a number of these models was observed albeit with large deviations. Although we do agree that further investigations on the Notch-EpCAM correlation are of interest, these data would considerably enlarge the size of the revised manuscript without, in our opinion, directly addressing the question raised by this reviewer.

4. If the authors choose to retain the metastasis experiments in a revised submission, then they should test the effect of withdrawing Dox after metastatic colonisation to favour proliferation following metastatic seeding? It would be interesting to know whether the timing of Zeb1 expression is important. Does conditional depletion of ZEB1 result in death of these EpCAM-low cells?

We agree on the importance of the in vivo metastasis experiments. However, these are complicated by a number of logistic and intrinsic limitations. The proposed experiments would imply large number of different animal models to be treated under different experimental conditions. Although technically feasible, these experiments are likely to take an extremely long time to be completed (up to 1 year, if not longer) and, due to the intrinsic variability of the outcome of orthotopic organoid transplantations, of uncertain outcome. In our opinion, the essential issue that these experiments should address is whether and how Zeb1 expression affects local invasion and distant metastasis formation in the AKP-Z orthotopic model. As shown in the original manuscript, while ZEB1 expression in the primary tumor does result in a substantial increase of the EpCAM^lo^ subpopulation of quasi-mesenchymal cells, long-term doxycycline administration increases only marginally the multiplicity of lung and liver metastases, likely to stem from the Zeb1-driven inhibition of MET, i.e. the reacquisition of epithelial features essential for metastatic colonization. However, to date our attempts to fine-tune dox administration so to enhance partial EMT and distant metastasis have not succeeded. This is partly due to the intrinsic variability of the orthotopic transplantation procedure but also to the limited number of ‘pulsed’ administrations that one can test.

Rather than removing the metastasis experiments altogether from the revised manuscript, we have moved the AKP-Z orthotopic transplantation data to primarily show that in vivo Zeb1 expression result in a substantial increase of the EpCAM^lo^ subpopulation in a cell line-independent experimental setup. For the sake of honesty, we have also retained the results of the effects on metastatic incidence upon long-term dox administration.5. In the AKP organoids, are the EpCAM-low cells also present and what is the endogenous expression status of ZEB1 in these?

Intestinal organoids (*enteroids*), when mutated in *Apc* or any other gene resulting in constitutive Wnt activation, or when cultured in Wnt3a-conditioned medium (or supplemented with the GSK3β-inhibitor Chiron), invariably grow as cystic *spheroids*, i.e. undifferentiated hollow 3D structures lined by a monolayer of *Lgr5*^low^ cells reminiscent of foetal intestinal progenitor cells. Only upon orthotopic transplantation of these spheroids into the cecum and the formation of the primary lesion, the organoid-derived cells acquire more pronounced epithelial features (EpCAM^hi^) and de-differentiate into EpCAM^lo^ upon *Zeb1*-induction. Moreover, there is hardly any endogenous Zeb1 expression in the AKP organoids (**Figure S1-5C**).

Why have the authors maintained the conditional expression of ZEB1 induced in the AKP-Z organoid transplantation experiments? This is likely driving the epithelial cells into a locked mesenchymal state – which is not compatible with the earlier observed 'plasticity' with the EpCAM-low cells in SW480 and HCT116 cells. Since EMT is generally believed to be essential for metastasis formation, the authors should consider temporal ZEB1 expression control in transplanted AKP-Z organoids if these experiments are to be retained in a revised submission.

See our reply to point #4. The identification of the ‘just-right’ experimental conditions for temporal Zeb1 expression in vivo to maximize metastasis induction (compared with continuous induction) is likely to take a considerable amount of time with uncertain outcomes due to the intrinsic variability of the orthotopic transplantation procedure. As stated before, in the revised manuscript. we have kept the transplantation experiments to show that in vivo Zeb1 expression result in a substantial increase of the EpCAM^lo^ subpopulation in a cell line-independent experimental setup, and focus on the RNAseq data and the newly implemented analyses, and their validation in patient-derived data sets. The admittedly marginal increase in lung and liver metastases is indeed explained by the Zeb1-driven inhibition of MET, i.e. the reacquisition of epithelial features essential for metastatic colonization.

6. In Figure 7, RNA sequencing identifies that Wnt signaling is likely enhanced in EpCAM-low cells. GSK inhibition induces the expression of ZEB1 (as known before), yet this works only in HCT116 and not in SW480 cells, which actually show an induction of Wnt signaling. The results seem to indicate that there is not just a mere enhancement of Wnt signaling and that other changes/pathways are required as well. Have the authors looked in other lines?

Please note that the induction of ZEB_1_ expression was observed in both cell lines and not only in SW480. This aside, the relationship between Zeb1 expression and Wnt signalling is not a straightforward one and we do agree on that additional pathways are likely to be involved. We looked into other colon cancer cell lines (mostly Apc- or β-catenin mutant and as such Wnt-on) merely to characterize the size of the EpCAM subpopulations (Source data 1) but have not looked into the differential levels of other pathways’ activation between EpCAM^lo^ and EpCAM^hi^ as this does not seem directly relevant for the ‘take-home’ message of the manuscript.

7. As Wnt signalling is proposed to be important in the establishment of the EpCam-low population, have the authors inhibited this pathway (either at the ligand level or though inhibiting b-catenin transcription) to confirm that the population is Wnt responsive?

This type of evidence was delivered in genetic fashion, i.e. by generating wild type, heterozygous, and homozygous β-catenin mutant clones of the HCT116 cell line (**Figure 3F**). Conversion of the mutant gene into its wild type version reduces the EpCAM^lo^ subpopulation from 29% to 4% of the total. If considered necessary, Wnt can be inhibited at the ligand level or by shRNA directed against β-catenin. This said, given the already provided evidence (including the central role of Zeb1, downstream of Wnt, in the maintenance of the EpCAM^lo^ subpopulation), the value of these additional experiments would be mainly incremental.

8. The scRNA sequencing seems to reflect EMT full and hybrid stages. The computational analysis is impressive and exciting, the potential trajectories offer a working model which could be experimentally tested by functional validation of the subgroups to finally pinpoint the cell populations with the highest cell plasticity. Most importantly, what defines cell plasticity at the molecular and cellular level? Is it Wnt signaling or something in addition (see point 7 above)? The reader is left without a clear answer. In the scRNA seq, in the populations that have increased EMT and EMT-gene expression, does this correlate to a Wnt/B-catenin signature at a single cell level?

What defines cell plasticity at the molecular and cellular level clearly represents a central question that we are trying to address using as experimental models colon cancer cell lines encompassing differentially plastic (and metastatic) subpopulations kept in equilibrium by stochastic state transitions. This should *a priori* be kept in mind taking into consideration that eliciting plasticity in vivo involves complex paracrine interactions between cancer cells located at the invasion front and their stromal microenvironment, which we believe are reproduced in cell-autonomous fashion in the cell lines.

In order to better address the reviewer’s question and further evaluate the correlation between Wnt and EMT at the single cell level, we have taken advantage of the Mesenchymal (Mes) and Epithelial (Epi) scores to rank the cells according to their level of EMT activation (**Figure 5C**) and correlate these with the activity of signalling pathways from the PID pathway database. From the N = 196 pathways encompassed in the PID database, Wnt signalling listed among those with highest correlation and showed significant increase in the EpCAM^lo^ population compared to the EpCAM^hi^ cells. Moreover, the overall “Wnt” score together with specific downstream target genes such as *AXIN2* and *DKK1*, showed positive correlation with the Mes-score (**Figure 5D**). Hence, given the intrinsic limits of our experimental approach, specific levels of canonical Wnt signaling are central to colon cancer cell plasticity and the manuscript accordingly focusses on this pathway. This does not of course imply that Wnt is the only pathway involved, especially in view of the role played by the tumor microenvironment in eliciting not only Wnt but also other synergistic pathways.

9. What is the mechanistic basis for the "further enhancement" of Wnt signaling? Is it the dose of Wnt signaling or is it the combination with other signaling pathways which cooperate with Wnt transcriptional control, such as Hippo or TGFb signaling? There could be a hint from the RNA sequencing data to distinguish these possibilities. Do the target gene lists change with the enhancement of Wnt signaling?

As it is likely to be the case for the majority of signalling pathways, Wnt is not a binary ON-OFF process but rather a dosage-dependent system where the level of the upstream signal is reflected by the downstream response not only in quantitative but also qualitative (i.e. different target genes) fashion. This is best illustrated by the so-called β-catenin paradox (*Curr Opin Cell Biol* 2007, 5:745-749): notwithstanding the loss of APC function (or oncogenic β-catenin activation) and the consequent constitutive activation of the Wnt signalling pathway, the majority of colon cancer cells do not show nuclear β-catenin accumulation with the only exception of those located along the invasive front. This shows that the original genetic defect is necessary but insufficient for full-blown pathway activation. The interaction with additional cues from the tumor microenvironment is likely to act synergistically to further enhance Wnt signalling levels. The latter is presumably the result of cooperation with other pathways as suggested by the reviewer and as supported by the scientific literature. As indicated above, our pathways analysis in correlation with EMT activity has indicated a number of candidate signalling routes likely to synergize with Wnt in eliciting plasticity. However, a more precise analysis of the key pathways such as Hippo or Tgf-β, is limited by the sequencing depth and the subset of genes whose activity is detectable by scRNAseq.

Next to synergistic pathways, incremental paracrine and autocrine Wnt stimulation may also underlie these observations. In our study, we have shown that when cultured in the presence of Wnt-conditioned medium or of Chiron, the *APC*- and β-catenin-mutant SW480 and HCT116 cell lines do respond by further increasing their intrinsically elevated level of Wnt activation and the EpCAM^lo^/Zeb1^+^ subpopulation (**Figure 3C-D**). Accordingly, it has also been shown that cancer cells already carrying Wnt activating genetic mutations secrete Wnt ligands in autocrine fashion to further enhance constitutive pathway activation (*Cancer Cell* 2004, 6:497-506).

10. Is the prognostic and predictive value for the gene signature only true for CMS4 CRCs or for all subtypes? Does the EpCAM-low signature and the signatures of the various EMT stages correlate with CMS subtypes, therapy resistance or clinical outcome? This is not really clear from the data presented.

As depicted in Figure 6, in the revised manuscript we developed a novel signature (CMS4_TC), obtained by selecting CMS4 genes exclusively expressed in epithelial cells, and compared with the CMS4_RF classifier. The CMS4_TC signature is preferentially expressed in tumors from CMS4 and the CMS1 subtype. Of note, Kaplan-Meier analysis showed that the CMS4_TC signature outperforms the CMS4_RF classifier when stratifying colon cancer patients according to relapse-free survival (Figure 6C,F), as it does for CMS1 and 3 patients (Figure S6-1).

11. The data depicted in Figure 10A and B are confusing and deserve better explanation. How is it possible that EpCAM-low and EpCAM-high sorted cells show overlapping single cell expression profiles upon t-sne plotting in particular for the SW480 cells? This is contradictory to the earlier claims of the authors in the manuscript that EpCAM-low cells have a more mesenchymal gene expression profile. Is there a difference between EpCAM protein expression and EpCAM mRNA expression? Regarding the heatmap from the EMT signature shown in figure 10B, which cell line is represented?

The tSNE of SW480 cells (Figure 5A and S5-1A) show overlap between the two subpopulations due to the fact that the genes with highest variance (e.g. RPL genes in the first principal component) are not differentially expressed between EpCAM^hi^ and EpCAM^lo^ cells (Figure S5-1B). In HCT116, these genes are differentially expressed across the population. However, EMT related variance is still capable of resolving the SW480 subpopulations as evidenced by the supervised tSNE (Figure 5B).

The heatmap in Figure 5C in the revised manuscript. displays both cell lines. The tSNE plots are labeled with the respective cell lines.

12. The authors link the gene expression signature of EpCAM-low cells with the colon cancer consensus molecular subtype CMS4 which have the worst relapse free and overall survival (Dienstmann R et al. 2017; 17, Nat Rev Cancer 79-92). There are multiple lines of evidence that the mesenchymal signature in CMS4 colon cancers is due to profound infiltration of stromal cells (CAFs, immune cells), extracellular matrix remodeling, TGF-β pathway activation and not the consequence of EMT in cancers cells (e.g. Calon et al. 2015; DOI: 10.1038/ng.3225). It is of course possible that a few epithelial cells in this inflammatory context are undergoing a partial EMT but there is little evidence and this likely will happen in a minority of cells. Therefore, the authors should revise the manuscript manuscript regarding (partial) EMT and the CMS4 and put their findings in a more critical context of what is already in the literature.

See also our reply to point #3. Indeed, many EMT-associated genes integral to the CMS4_RF classifier are highly expressed in stromal cells. To elucidate the role of EMT in tumor epithelial cells, we derived the CMS4_TC signature by selecting genes that correlate with the CMS4_RF signature within the epithelial fraction. Evaluation of both signature in all cell types (from Lee et al. Nat Genet 2020, 52:594–603) showed that CMS4_TC is specific for tumor cells (Figure 6D) while CMS4_RF shows highest expression in stromal cells (Figure 6A); also, CMS4_TC shows association with the CMS1 and CMS4 molecular subtypes (Figure 6G) and outperforms other classifiers (including CMS4_RF) in stratifying CMS patients other than 4 (CMS1 and 3) (Figure 6C,F and S6-1). Using the CMS4_TC signature, quasi-mesenchymal epithelial cells were found to have elevated expression levels of non-classical EMT genes. We then evaluated the marker genes from the mesenchymal-like epithelial cells in our sc- and bulk-RNAseq data from the two cell lines (Figure 5C). Notably, expression of SPARC appeared to be higher in the EpCAMlo cells when compared to the EpCAM^hi^ population, and peaked in between the EMT axis, indicative of cells in partial EMT (Figure 5D). Accordingly, cells with the highest hybrid EMT score showed increased plastic potential as evidenced by the opposing arrows on the RNA velocity plot (Figure 5EG). In view of the role of SPARC in modifying the ECM and eliciting EMT through the upregulation of other genes like FN1 (fibronectin), MMP7, CD44 and others, it is likely that the pEMT state is earmarked by the coordinated expression of a subset of genes favoring local invasion through, for example, collective cell migration. Accordingly, overexpression of SPARC in the original HCT116 and SW480 cell lines results in the up- and downregulation of M- and E-markers respectively, and increase their invasive capacity (Figure 5H).

Taken together, while classical EMT genes may be scarcely and occasionally expressed in primary bona fide colon cancers, specific subsets exist featuring a quasi-mesenchymal state that correlates with poor survival and is preferentially observed not only in CMS4 tumors but also in a CMS1 and CMS3 cases. Of note, these epithelial cancer cells with mesenchymal features express increased levels of SPARC, a gene likely to earmark partial EMT in a subset of the cases.